# Effects of Salt Soaking Treatment on the Deodorization of Beef Liver and the Flavor Formation of Beef Liver Steak

**DOI:** 10.3390/foods12203877

**Published:** 2023-10-23

**Authors:** Yufeng Duan, Ziqi Liu, Dan Deng, Li Zhang, Qunli Yu, Guoyuan Ma, Xiaotong Ma, Zhaobin Guo, Cheng Chen, Long He

**Affiliations:** College of Food Science and Engineering, Gansu Agricultural University, Lanzhou 730070, China; 17899318727@163.com (Y.D.); 2483918511@163.com (Z.L.); 17899316114@163.com (D.D.); yuqunlihl@163.com (Q.Y.); maguoyuangsau@163.com (G.M.); maxiaotong17@163.com (X.M.); guozhb007@163.com (Z.G.); chenchengmlj@163.com (C.C.); 18215284414@163.com (L.H.)

**Keywords:** GC–IMS, volatile flavor compounds, meat by–products, salting–out effect

## Abstract

In this study, based on the evaluation of fishy value and sensory evaluation, this study determined that soaking in a 1% salt solution for 60 min had a significant impact on the deodorization of beef liver (*p* < 0.05). The results showed that salt infiltration promoted the release of fishy substances, improving the edible and processing performance of beef liver. The identification of flavor compounds in raw and roasted beef liver via GC–IMS implies that (E)–2–octenal–M, (E)–3–penten–2–one–M, ethyl acetate–M, ethyl acetate–D, and methanethiol are closely related to improving the flavor of beef liver; among them, (E)–2–octenal–M, (E)–3–penten–2–one–M, and methanethiol can cause beef liver odor, while nonanal–M, octanal–M, benzene acetaldehyde, n–hexanol–D, butyl propanoate–M, heptanal–D, heptanal–M, and 3–methylthiopropanal–M had significant effects on the flavor formation of beef liver steak. The determination of reducing sugars revealed that salt soaking had no significant effect on the reducing sugar content of beef liver, and the beef liver steak was significantly reduced (*p* < 0.05), proving that reducing sugars promoted the formation of beef liver steak flavor under roasting conditions. Fatty acid determination revealed that salt soaking significantly reduced the content of polyunsaturated fatty acids in beef liver (*p* < 0.05), promoting the process of fat degradation and volatile flavor production in the beef liver steak. Salt plays a prominent role in salting–out and osmosis during deodorization and flavor improvement. Through controlling important biochemical and enzymatic reactions, the release of flavor substances in a food matrix was increased, and a good deodorization effect was achieved, which lays a foundation for further research on the deodorization of beef liver and the flavor of beef liver steak.

## 1. Introduction

Meat by–products account for nearly 60–70% of slaughtered carcasses, of which nearly 70% are edible [1]. With the increasing global demand for high–protein food products, one of the main challenges facing the meat by–products industry is how to increase the value of by–products in order to improve the profitability of the industry, enable environmentally friendly production, and provide innovative and nutritious products. Among edible offal, beef liver has a good taste and similar protein content to beef, and it is rich in glycogen and minerals and low–fat content. It is of high medicinal value and rich in fat–soluble vitamins, L–carnitine, glutathione, and other active substances, which have important physiological and healthcare functions. As a high–quality protein food, it is very beneficial to human health [2,3,4], and it can be directly cooked or processed into various special foods [5]. In Western countries, it is often processed into beef liver sausage, canned beef liver sauce, etc., [6,7]. As a Western dish, beef liver steak is mainly made from beef liver through pretreatment, deodorization, roasting, packaging, and other processes. Currently, there is no report on the quality formation of beef liver steak and industrial technology research and development. The main consumption mode of Chinese beef liver is fresh food sales, and the processed products comprise mainly pickled liver products and soy sauce liver products [8,9,10]. However, the bad flavor of liver not only affects its cooking and consumption by families but also has certain negative effects on its processing and sales. Therefore, the deodorization of beef liver is a key issue in order to increase the acceptability of beef liver and improve the odor of beef liver products (e.g., beef liver steak).

Deodorization technology includes chemical deodorization, biological deodorization, and physical deodorization [11]. The chemical deodorization method usually uses a combination of acid and alkali to react with the odor compounds to control odor. However, chemical residues and possibly harmful substances have attracted wide public attention and are not suitable for use in food processing processes [12]. Bio–deodorization is a method of deodorization using microbial fermentation techniques [13]. These deodorization methods comprise mainly adsorption, but they may produce additional odors, seriously affecting the physical and chemical properties of the products, often requiring special treatment methods, and they are not easy to control [14]. Salt is used as a substitute for yeast extract, monosodium glutamate, lactate, and spices in the modern industry to achieve flavor enhancement or masking effects [15]. With the physical deodorization method, salt solutions promote the formation of flavor and savory characteristics by controlling important biochemical and enzymatic reactions, and they have a desalting effect that increases the release of flavor compounds from the food matrix [16,17]. As a low–cost deodorization method, salt can achieve a better deodorization effect.

The main purposes of this study were (1) the identification of flavor compounds in raw and grilled porcini steaks using gas chromatography–mass spectrometry (GC–IMS), (2) based on the evaluation of fishy value and sensory evaluation, the analysis of aromatic active compounds of beef liver and beef liver steak under different soaking conditions and the determination of compounds causing a bad flavor, (3) and the revelation of the effects of salt soaking on the deodorization of beef liver and the formation of beef liver steak flavor in the processes of salting–out, enzymatic reaction, and lipid oxidation. This study provides a new way to determine the deodorization of beef liver and improve the flavor of beef liver steak.

## 2. Materials and Methods

### 2.1. Materials

Beef liver was obtained from Kangmei Modern Farming and Animal Husbandry Group Co. Six cows with good growth and development and similar physiques that met the quarantine standard. The average age of the cattle was between 1.5 and 2 years. Their average weight was about 330 kg, and the six cattle were from the same feeding batch. All cattle were slaughtered in the local slaughterhouse. Before slaughter, all test cattle were fasted for 24 h and slaughtered after water was forbidden for 2 h. Immediately after slaughter, the beef liver was removed, the surface of the liver was washed with water, the fascia on the surface of the liver was removed, and the liver was cut into cubes of uniform weight and frozen at −18 °C until it was analyzed. It was thawed at 0–4 °C for 10–12 h before each use. The salt purchased from China Salt Gansu Salt Industry (Group) Company Limited, Chengguan District, Lanzhou City, Gansu Province, China.

### 2.2. Sample Processing

The soaking design for the beef liver was as follows: at 4 °C, the beef liver was soaked in a 1.0% salt solution and distilled water for 30, 60, and 90 min. The control group comprised beef liver soaked in distilled water and untreated beef liver. The baking design of the beef liver steak was as follows: the oven was preheated for 3 min in addition to the soaking process, a baking tray was brushed with cooking oil, the marinated beef liver steaks were placed on the tray, undergo the baking process (200 °C, 7 min). The control group consisted of roasted beef liver steak that was soaked in distilled water and not roasted. The experimentally treated samples were numbered as shown in Table 1. The experimental design for the effect of salt soaking treatment on the deodorization of beef liver and the flavor formation of beef liver steak is shown in Figure 1.

### 2.3. Sensory Evaluation

A training program of three 20 min sessions was completed before the evaluation. Twelve volunteers (6 female and 6 male, aged 22–45 y, including both graduate students and faculty members from the Gansu Agricultural University of China, Lanzhou, Gansu, China) who were willing to evaluate the sample were selected for training using questionnaires. The samples of beef liver were taken and put into 10 mL sense–measuring cups. The sense assessors sniffed them quickly, and the intensity of the smell was evaluated using a 5–point system [18,19]. There were 5 levels according to the degree of odor (0 < no odor or slight odor ≤ 1; 1 < slight odor ≤ 2; 2 < moderate odor ≤ 3; 3 < heavy odor ≤ 4; and 4 < very heavy odor ≤ 5).

The beef liver steak samples were tasted, and a sensory evaluation was carried out using the method previously described with some modifications [20]. People can accurately describe the texture, state of organization, color, taste, and overall acceptability of food by looking at it, touching it with chopsticks, smelling it, and tasting it. There were four levels of sensory evaluation: 20–16, 15–11, 10–6, and 5–0. The sensory evaluation scores were averaged.

### 2.4. Gas Chromatography–Ion Mobility Spectrometry (GC–IMS) Analysis

GC–IMS (Flavourspec, G.A.S. Instrument, Dortmund, Germany) and Column MXT–5 (weak polarity, 15 m × 0.53 mm × 1 μm) (REXTEC, Bellefonte, PA, USA) were used to analyze the volatile compounds in the samples. The samples were thawed in the refrigerator at 4 °C overnight (12 h) before the experiment. The samples were analyzed using a GC–IMS instrument with minor modifications, as described [21]. The samples of beef liver and steak under different treatment conditions of 3 g were accurately weighed and put it into a 20 mL headspace (HS) vial with a magnetic screw seal cap. Then, they were incubated at 60 °C and 500 rpm in headspace at the incubation time of 15 min, and after incubation, injection volume was 500 uL a 500 uL injection volume was used. The column temperature was maintained at 60 °C, the drift tube temperature was 45 °C, and the drift gas flow rate was set to a constant flow rate of 150 mL/min. A nitrogen carrier gas (99.999% purity) was used, the gas chromatographic column flow rate was set to 2 mL/min and kept for 2 min, and then, it was linearly increased to 100 mL/min within 20 min. Each sample was collected three times as a parallel sample.

### 2.5. Determination of Fatty Acid Content

Fatty acid analysis was performed following Dominique Gruffat et al., [22]. Before GC analysis, the fatty acids in the sample were first released via hydrolysis and esterified to fatty acid methyl ester. Then, 0.4 mL of KOH and 3 mL of anhydrous methanol were added, and the test tube was bathed in 55 °C water for 90 min and shaken for 5 s every 20 min. After the water bath, the test tube was removed and cooled to room temperature. Then, 0.33 mL of H2SO4 solution was added, and the test tube was bathed in 55 °C water for 90 min with shaking every 20 min for 5 s. It was then cooled to room temperature, and 1.7 mL of n–hexane (with methyl ester) was added and homogenized for 5 min using a homogenizer at a speed of 5000 RPM. The methylated samples were filtered into a 0.22 µm high–performance liquid chromatography flask (Agilent, Santa Clara, CA, USA) for chromatographic analysis. The chromatographic analysis and measurement parameters were as follows: The chromatographic column was an HP–5MS capillary column (30 m × 0.25 mm × 0.25 µm in film thickness, Agilent, USA), the column temperature was 150 for 3 min, and then it was increased to 180 at a rate of 2.5/min and maintained for 5 min. Then, the temperature was increased to 220 at a rate of 2.5/min, and the carrier gas rate was maintained at 0.7 mL/min for 25 min. The fatty acid composition was identified using the mass spectrometry database (NIST Library, Mass Spectrometry Retrieval Program, version 5.0, USA).

### 2.6. Determination of Reducing Sugar Content

The beef liver of each treatment group weighed about 1 g, and a little quartz sand and 1 mL 10% trichloroacetic acid solution was added before it was ground it to a mealy shape. Then, 2 mL of 5% trichloroacetic acid was added to continue grinding for a moment to form a uniform, mealy slurry. It was transferred into a centrifugal tube and centrifuged at 2500 r/min for 10 min. The supernatant was placed into another centrifuge tube, an equal volume of 95% ethanol was added, and it was mixed well and allowed to stand for a moment to precipitate glycogen in flocculent form. Then, it was put it into the centrifuge at 2500 r/min for centrifugation for 10 min. The supernatant was discarded, and the centrifuge tube was inverted on the filter paper. Then, the method was determined via direct titration, according to GB 5009.7-2016.

### 2.7. Statistical Analysis

The experiments were performed in triplicate, and the results were represented by mean values ± standard deviations. The SPSS 22 and Origin 2021 software were used for the experimental data analysis. The significance of the difference was analyzed via Duncan’s multiple comparison method, and *p* < 0.05 indicated a significant difference. The characteristic flavour substances were qualitatively analyzed by the software LAV in GC-IMS and the databases of NIST2014 and IMS in the GC-IMS Library. The GC–IMS spectra of the samples were compared using a plug–in in LAV Reporter, and the GC–IMS fingerprints were compared using the plug–in Plot Gallery.

## 3. Results and Discussion

### 3.1. Sensory Evaluation

#### 3.1.1. The Fishy Value of Raw Beef Liver after Soaking

Sensory assessment is a validated and reliable technique [20], and the change in the odor value of raw beef liver after immersion can directly reflect the deodorization effect of raw beef liver under different immersion conditions. Figure 2a shows the change in fishy values of the raw beef liver before and after soaking in distilled water and table salt compared to the unsoaked raw beef liver. After soaking in distilled water for 30, 60, and 90 min, the fishy value of the raw beef liver decreased from 5 to about 3, but there was no significant difference between groups (*p* > 0.05). The results showed that soaking in distilled water reduced the fishy value of beef to some extent. After soaking in a 1% salt solution for 30 min, 60 min, and 90 min, during which the deodorization value of 1% salt for 60 min was the lowest at about 1, the deodorization effect was the best, and there was a significant difference (*p* < 0.05). A possible reason is that salt osmosis promotes the release of fishy substances, but salt osmosis tends to saturate as the time of soaking raw beef liver increases.

#### 3.1.2. Sensory Score of Roast Beef Liver Steak

Traditionally, product grades are provided via sensory evaluation, which provides a comprehensive measure of the acceptability and immediate target attributes of a food product, such as color, aroma, taste, and texture [20], Figure 2b is a radar image of the sensory evaluation of roast beef liver steak under different preprocessing conditions. According to five sensory evaluation indexes—texture, tissue state, color and luster, flavor, and overall acceptability—it can be seen that there are obvious differences in the sensory quality of roast beef liver steak with different pretreatments. Under the condition of soaking in distilled water, the texture of the beef liver steak for 60 min > 30 min and > 90 min, the color, luster and flavor for 60 min > 90 min and > 30 min, the tissue condition and overall acceptability of 60 min was the best, and there was no significant difference between 30 min and 90 min. And the overall score of the beef liver steak after soaking in distilled water for 60 min was higher than that for 30 min and 90 min. It was shown that the soaking effect of distilled water improved the overall score of the beef liver steaks, which was consistent with the overall decrease in the fishy value of the raw beef liver after soaking with distilled water, but a longer or shorter soaking time affects the organoleptic quality of the steaks. Under the condition of 1% salt soaking, the texture, tissue state, color and luster, flavor and overall acceptability of the beef liver steak were the highest at 60 min of soaking (that is, the sensory quality was the best), followed by 30 min of soaking and finally, for 90 min, the texture, texture, color, flavor, and overall acceptability were not ideal. From the radar map, it can be seen intuitively that, after salt soaking, the roast beef liver steak’s overall score was generally better than that of the distilled–water–soaked beef liver steak. Salt has the effect of increasing the fresh taste of flavor substances. The combination of amino acids in raw food materials and sodium in salt will form sodium amino acids with a strong umami taste; this phenomenon indicates that salt soaking can impart a better flavor and taste to beef liver to a certain extent and improve the edible and processing properties of beef liver. Salt pickling has been applied to inhibit the growth of Gram–negative bacteria and inhibit enzyme–related chemical reactions in meat products by reducing water activity [23]. In traditional manufacturing, salt processing can produce unique flavored products that better cater to consumer needs. Meanwhile, this study also found that salt water can lead to a series of mass transfer processes, including Na^+^ or Cl^–^ diffusion and water seepage [24]. It may promote the release of fishy substances. However, as the soaking time of raw beef liver increases, the salt permeation tends to be saturated; saltwater immersion can affect the commercial quality of fish products, including water retention capacity, texture properties, etc. [25,26]. Therefore, it is necessary to control the soaking time and concentration to achieve dual guarantees of beef liver quality and deodorization.

### 3.2. GC–IMS Analysis of Volatile Compounds

Headspace–gas chromatography–ion mobility spectrometry (HS–GC–IMS) is an analytical technique for the detection of trace gases and the characterization of chemical ion species based on differences in the migration rates of different gas–phase ions in an electric field. This detection technique uses gas chromatography retention time (GC) and ion mobility spectroscopy (IMS) drift time to achieve the two–dimensional separation of substances [27]. Figure 3a shows the comparison and difference spectra of the volatile components of raw beef liver under different soaking methods, and Figure 3b shows the comparison and difference spectra of the volatile components of roast beef liver steak under different soaking methods. These two spectra represent all the volatile compounds in beef liver and beef liver steak, respectively. The red vertical line on the left represents the reaction ion peaks, and each point represents the volatile organic compounds in the sample. In Figure 3a,b, x– and y– respectively represent the ion migration time for identification and the retention time of the gas chromatograph. Some differences were found in the peak signal intensity of each sample, indicating differences in the content of volatile flavoring substances in different samples. Most of the signals occurred during 100–300 s of hold time and 1.0–1.75 s of drift time. The color represents the signal strength of the substance. White indicates low strength, red indicates high strength, and the strength increases as the color deepens [28]. As can be seen from Figure 3a, the volatile flavor components of beef liver under different soaking conditions were similar, but the signal intensity was slightly different; with the treatment group soaked in distilled water compared with the control group without soaking, the signal intensity of samples after soaking for 30 min and 90 min did not change obviously, and the red spots of volatile substances increased after soaking for 60 min. The amount of volatile substances released from part of the raw beef liver increased first and then decreased, and the signal was strongest at 60 min. The same phenomenon was consistent in the salt soaking deodorization group, in which 1.0% salt was used for soaking for 60 min, and the difference in the flavoring substance content in the beef liver was the greatest compared with the control group. When the retention time was 100–200 s, the blue spot was greater, and the signal intensity of most compounds disappeared or weakened. In contrast, the enhancement of some signals indicated that the concentration of these compounds changed significantly under the condition of 60 min immersion in a 1.0% salt solution. This may be due to the obvious migration of flavor components after soaking in a 1.0% salt solution for 60 min, resulting in a significant increase in the content of some flavor components and a decrease in the fishy value. Figure 3b shows that the salt group (KC–30, KC–60, and KC–90) had more volatile flavor compounds than the distilled water group (KB–30, KB–60, and KB–90); this may have been due to the addition of salt to promote the release of certain volatile flavor components, which may contribute to the flavor improvement of steak.

Topographic plots can visually show the change trends of volatile components. However, it is hard to make correct judgments using the closely connected substances on the map. This problem can be solved well using a fingerprint. According to the peak signal, the beef liver fingerprint was formed. In the fingerprint, each row shows all the signal peaks of a sample, and each column shows the same volatile compound in a different sample, and its color represents the content of the volatile compound. The brighter the color, the higher the content. The dynamic changes of volatile compounds in different samples can be compared via fingerprints [21]. Figure 3c shows the fingerprints of raw and roasted beef liver steaks treated using different soaking methods, in which 70 volatile compounds were identified and determined in the samples of beef liver; they were mainly aldehydes, ketones, alcohols, and esters, among which there were 16 aldehydes, 10 ketones, 9 alcohols, 6 esters, and 2 hydrocarbons, and there were 1 furan, 1 acid, and unknown components. The content of volatile organic compounds in the roast beef liver steak was significantly different from that in the raw beef liver. Some of the colors were markedly darker, increasing the concentration of volatile substances. The corresponding substances were 3–hydroxybutan–2–one–M, (E)–3–penten–2–one–M, butyl acetate, 2–heptanone–D, 2–heptanone–M, 3–methylbutanal–D, butanal, pentanal–M, heptanal–D, heptanal–M, hexanal–D, hexanal–M, (E)–2–pentenal–M, 3–methylbutan–1–ol, n–hexanol–D, n–hexanol–M, pentan–1–ol–D, pentan–1–ol–M, (E)–2–hexen–1–ol–D, (E)–2–hexen–1–ol–M, oct–1–en–3–ol–D, oct–1–en–3–ol–M, and 2–ethyl–1–hexanol; some colors are noticeably lighter, corresponding to lower concentrations of hexanoic acid, 3–hydroxybutan–2–one–D, 1–octen–3–one–D, 1–octen–3–one–M, benzaldehyde–D, and benzaldehyde–M. By comparing the point intensities of the volatile flavor compounds in raw beef liver and beef liver steak under different soaking conditions, the differences in volatile flavor compounds in the raw beef liver and the beef liver steak were determined. 

### 3.3. Volatile Flavor Compounds in Beef Liver

#### 3.3.1. Identification of Volatile Flavor Compounds in Soaked Raw Beef Liver

The volatile organic compounds of beef liver under all the different soaking conditions are listed in Table 2. Most of these compounds come in two forms: monomers and dimers. In order to further analyze the odor substances of beef liver, the relative content of volatile organic compounds was calculated according to the peak area, and histogram analysis was carried out for the substances with significant differences among the treatment groups; among them, aldehydes were the most widely detected compounds, with lower sensory thresholds that often contribute to the generation of ideal aromas with grassy and fatty odors [29,30]. As shown in Figure 4a, substances with significant differences in the relative content of aldehydes between treatments included hexanal–D, hexanal–M, (E)–2–pentenal–D, 3–methylbutanal–D, 3–methylbutanal–M, and 2–methylbutanal–D. Of the six substances detected after immersion in a 1.0% salt solution for 60 min, for hexanal–D, hexanal–M, 3–methylbutanal–D, 3–methylbutanal–M, and 2–methylbutanal–D, the relative contents of the five aldehydes were significantly higher than those of the other treatments (*p* < 0.05); however, the relative content of 3–methylbutanal–D, 3–methylbutanal–M, and 2–methylbutanal–D was lower, while the thresholds of hexanal–D and hexanal–M were higher. According to the results of the sensory evaluation, the best deodorization effect was obtained when the cow liver was soaked in a 1% salt solution for 60 min. The relative content of (E)–2–pentenal–D after soaking in a 1.0% salt solution for 60 min was significantly lower than that of the other treatment groups (*p* < 0.05). (E)–2–pentenal–D is an active flavor compound produced via lipid oxidation with fat flavor. It has an important effect on food flavor quality, and its relative content decreases after 60 min soaking in a 1.0% salt solution. Therefore, (E)–2–pentenal–D may be one of the components that cause the odor of beef liver.

Most ketones have a high threshold and contribute little to flavor properties, while some ketones are important intermediates of heterocyclic compound formation and play important roles in flavor formation [31,32]. Figure 4b shows that, among the ketone compounds detected, for 2–butanone–D, 3–hydroxybutan–2–D, (E)–3–penten–2–one–M, 2–butanone–M, 3–pentaone–M, 2–heptanone–M, 2,3–hexanedione–M, and 3–hydroxybutan–2–one–M, the relative contents of these eight ketones in the treatment groups were significantly different (*p* < 0.05). After soaking in a 1.0% salt solution for 60 min, the relative contents of 2–butanone–D and 3–hydroxybutan–2–one–M increased significantly compared with the other treatments; because of high thresholds, 2–butanone–D and 3–hydroxybutan–2–one–M little effect on the flavor contribution of bovine liver. The relative contents of 3–hydroxybutan–2–D and (E)–3–penten–2–one–M were significantly lower than those of the other treatments (*p* < 0.05). 3–hydroxybutan–2–D has a milky flavor, so the relative content of (E)–3–penten–2–one–M decreased after soaking in a 1.0% salt solution for 60 min, and it may be one of the substances that causes the odor of beef liver.

Figure 4c shows that, in esters, the relative contents of ethyl acetate–D and ethyl acetate–M were significantly different between the treatment groups (*p* < 0.05). After soaking in a 1.0% salt solution for 60 min, ethyl acetate–D and ethyl acetate–M significantly decreased compared with the other treatments, and all of them had a wine–like aroma. Because of their low volatility, they had only a weak regulative effect on the deodorization as a whole.

#### 3.3.2. Identification of Flavor Compounds in Beef Liver Steak under Different Soaking Conditions

In the roast beef liver steak, aldehydes such as nonanal–M, nonanal–D, octanal–M, octanal–D, benzaldehyde–M, benzaldehyde–D, benzene acetaldehyde, hexanal–D, heptanal–M, heptanal–D, 3–methylthiopropanal–M, pentanal–M, pentanal–D, butanal, (E)–2–pentenal–M, 2–methylbutanal–M, 2–methylbutanal–D, 3–methylbutanal–M, and 3–methylbutanal–D were detected. These aldehydes showed significant differences compared to the other treatment groups under the condition of soaking in a 1% salt solution for 60 min (Table 3), among which nonanal–M and nonanal–D had a sweet orange flavor, octanal–M, octanal–D, heptanal–M, and heptanal–D all had a fatty flavor, both benzaldehyde–M and benzaldehyde–D had an almond aroma, benzene acetaldehyde had a floral aroma, hexanal–D had a grassy aroma, 3–methylthiopropanal–M had a brothy aroma, butanal had an elegant fragrance, and (E)–2–pentenal–M, 2–Methylbutanal–M, 2–Methylbutanal–D, 3–methylbutanal–M, and 3–methylbutanal–D had a fruity aroma. These aldehydes can, to some extent, help form the flavor of roasted beef liver steak soaked in a 1% salt solution for 60 min.

Ketones derived from the Maillard reaction and fat degradation [30,33,34,35], According to Figure 5a, with 3–pentanone–M, 3–pentanone–D, 3–hydroxybutan–2–one–M, 2,3–hexanedione–M, (E)–3–penten–2–one–M, (E)–3–penten–2–one–D, and 2–heptanone–M. After 60 min of treatment with a 1.0% salt solution, the relative contents of these ketones were significantly different from those of the other treatment groups. The relative contents of 2–heptanone–M, 3–pentanone–M, and 3–pentanone–D were significantly lower than those of the other treatments (*p* < 0.05), while the contents of 3–hydroxybutan–2–one–M, (E)–3–penten–2–one–D, 2,3–hexanedione–M, and (E)–3–penten–2–one–M were significantly higher than those of the other treatments (*p* < 0.05), but the ketones had little effect on the flavor of the roast beef liver steak because of their high universal threshold.

In the esters (Figure 5b), beef liver steak was roasted after soaking in a 1.0% salt solution for 60 min, and the relative contents of ethyl acetate–D and ethyl acetate–M were significantly lower than in the other treatment groups. The relative contents of butyl acetate and butyl propionate–D were significantly higher than those of the other treatments (*p* < 0.05); however, the relative contents of these esters in the roast beef liver steak were lower. Both ethyl acetate–D and ethyl acetate–M have very high butyl acetate thresholds and a wine–like aroma, while both butyl acetate and butyl propionate–D have a fruity aroma and a high threshold [36]. Therefore, soaking ethyl acetate–M, ethyl acetate–D, butyl acetate, and butyl propionate–D in a 1.0% salt solution for 60 min has a positive effect on the flavor formation of roasted beef liver steak, but the effect is relatively small.

Methanethiol, oct–1–en–3–ol–M, 1–butanol–M, (E)–2–hexen–1–ol–M, (E)–2–hexen–1–ol–D, 3–methylbutan–1–ol, pentan–1–ol–M, and pentan–1–ol–D were analyzed according to Figure 5c. The relative content of methanethiol and oct–1–en–3–ol–M was high. With the roasting of beef liver steak, more sulfur compounds are produced via the thermal degradation of thiamine, which is helpful in producing the basic taste of meat [35]. The content of oct–1–en–3–ol–M is often accompanied by a fruity flavor.

Among the other compounds, as Figure 5d shows, despite significant differences between the treatment groups, the relative amounts of hexanoic acid, 2–pentyl furan, and styrene were low. Among them, the threshold value of styrene was higher, and hexanoic acid has a strong flavor. So, caproic acid, 2–pentylfuran, and styrene may have little influence on the flavor of roast beef liver steak after soaking in a 1.0% salt solution for 60 min.

### 3.4. Principal Component Analysis (PCA) and Cluster of Volatile Compounds

To visualize the changes in the volatile components of beef liver and beef liver steak under different soaking methods, differences in the volatile components were highlighted using PCA, and heat maps of the volatile components were plotted. The cumulative contribution of the principal component of the flavor of bovine liver after soaking was 88.10%, as per Figure 6a, which better characterizes the original data. Among them, the characteristic volatile flavors of samples A, B–30, C–30, and B–90 in the purple circle of the left half of the axis sample of PC1 were similar, including ethyl acetate, methanethiol, (E)–3–penten–2–one–M, (E)–3–penten–2–one–D, etc. At the same time, observing the principal component area map on the right showed that the samples soaked for 60 min and soaked for 90 min were approximately located on both sides of PC2, respectively, in the purple circle. Under different soaking conditions, the contribution rates of roasted beef liver steak PC1 and PC2 were 45.7% and 30.6%, respectively (Figure 6b). The roasted beef liver steak under different treatment methods can be distinguished based on the distribution of volatile flavors, as shown in the orange coil. The aggregation of samples such as KB–30, KB–60, and KB–90 on the left side of PC1 proves that, with the same soaking solution treatment, the main flavor substances of roasted beef liver steak soaked in distilled water are similar, with KC–30 on the right side of PC2. The main difference between KC–60 and KC–90, which are located on the upper and lower sides, respectively, lies in the differences in flavor substances, such as butyl propanoate–D, butyl acetate, ethyl acetate–M, 3–pentanone–M, 3–hydroxybutan–2–one–M, 2,3–hexanedione–M, (E)–3–penten–2–one–M, (E)–3–penten–2–one–D, 2–heptanone–M, etc., which is consistent with the results of the cluster analysis.

In the cluster analysis diagram, the samples and volatile flavor compounds were constructed on the upper and left sides, respectively. The beef liver samples were divided into two groups under different soaking conditions (Figure 6c), one consisting of samples soaked for 60 min in a 1% salt solution for 90 min, and the rest were counted as the other group. The effect of the soaking time and treatment methods on the volatile flavor components can be further divided according to the clustering results. The beef liver steaks under different soaking conditions could be divided into two groups according to the treatment method (Figure 6d).

### 3.5. Changes in the Fatty Acid Content of Beef Liver and Beef Liver Steak under Different Soaking Conditions

Free fatty acids are important flavor precursors in meat products that are released during meat processing through lipid hydrolysis; the main volatile flavor compounds (aldehydes, ketones, alcohols, etc.) in processing are mainly derived from the oxidative degradation of fatty acids [37,38,39,40]. Table 4 shows the distribution of fatty acids in beef liver and beef liver steak under different soaking conditions. There were 16 fatty acids detected in the beef liver, and the fatty acid contents of the beef liver were different. Combining the heat maps of the fatty acid content in the beef liver and the beef liver under different soaking conditions (Figure 7a), it can be seen that the overall difference in the fatty acid content of the beef liver soaked in distilled water (B–30, B–60, and B–90) was not significant compared to the untreated beef liver. The content of polyunsaturated fatty acid in the beef liver soaked in a 1% salt solution (C–30, C–60, and C–90) was significantly lower than that of the beef liver soaked in distilled water (B–30, B–60, and B–90) and the beef liver that was not soaked (*p* < 0.05). When soaked in a 1% salt solution for 60 min, the relative contents of C12:0, C14:1, C16:0, C15:0, C20:2, and C20:3n6 were higher than those of the other treatment groups. It was speculated that this phenomenon might have been due to the interference of salt content and soaking time on the activity of some fat hydrolase, which affected the release of volatile flavor substances, reduced the odor value of the beef liver, and achieved a certain deodorization effect. More than half of the volatile compounds in cooked meat are produced via lipid oxidation and degradation. In the roasting process, the overall content of the beef liver steak was significantly higher than that of the raw beef liver for C14:1, C16:1, C17:1, C20:0, C20:1, and C20:2, and the polyunsaturated fatty acid was significantly reduced (*p* < 0.05). Salt added to meat can inhibit catalase, superoxide dismutase, glutathione peroxidase, and other antioxidant enzymes, thus accelerating the oxidation process, in which the thermal oxidation of unsaturated fatty acids is the main source of aldehydes and other volatile compounds [41]. Therefore, it has a certain improvement effect on the formation of beef liver flavor.

### 3.6. Changes in Reducing Sugar Content in Beef Liver and Beef Liver Steak under Different Soaking Conditions

Figure 7b shows the changes in the reducing sugar content in the raw beef liver and the beef liver steak under different soaking conditions. The content of the reducing sugar in the raw beef liver was about 5 g/100 g. Compared with the raw beef liver soaked in distilled water (B–30, B–60, and B–90) and the untreated beef liver (A), there was no significant difference between the treatment groups (*p* < 0.05). The sodium chloride immersion treatment had a relatively small impact on the content of reducing sugar in the raw beef liver. Under different soaking conditions, the content of reducing sugar in the raw beef liver steak dropped to 2.8 g/100 g, 2.5 g/100 g lower than that of the raw beef liver, and the content of reducing sugar in the roasted beef liver steak soaked in distilled water (KB–30, KB–60, and KB–90) and the roasted beef liver steak soaked in a 1% salt solution (KC–30, KC–60, and KC–90) also demonstrated the same phenomenon, which was due to the intensification of the Maillard reaction and the caramelization reaction caused by a high–temperature treatment [38]. As a substrate, reducing sugar contributes to the flavor and color formation of beef liver steak. The content of reducing sugar in the beef liver steak (KC–30, KC–60, and KC–90) was the lowest after soaking in a 1.0% sodium chloride solution, and its content was significantly lower than that of the other treatment groups (*p* < 0.05). Volatile sulfur compounds generated via the Maillard reaction are characteristic aroma substances, which are usually generated from the reaction of cysteine or glutathione with reducing sugar (such as ribose and glucose in meat) [39]. Therefore, the reduction in the reducing sugar content in bovine liver after soaking in sodium chloride may be attributed to the fact that sodium chloride promotes the occurrence of thermal reactions such as the Maillard reaction in a sense, which is of great significance to the formation of the flavor of beef liver. However, the mechanism of sodium chloride’s impact on the flavor of a heat reaction still needs to be further explored in combination with the impact of amino acids in the Maillard reaction.

## 4. Conclusions

This study focused on the method of simple salt deodorization and compared the effects of soaking in a 1% salt solution on beef liver and beef liver steak with non–deodorization and distilled–water soaking deodorization. It clarified the important significance of salt soaking in beef liver and beef liver excretion, and it used GC–IMS to analyze the aromatic active ingredients of beef liver and beef liver steak under different soaking conditions. It identified the important substances that contribute to the deodorization and flavor formation of beef liver, providing new ideas about the deodorization and utilization of beef by–products.

## Figures and Tables

**Figure 1 foods-12-03877-f001:**
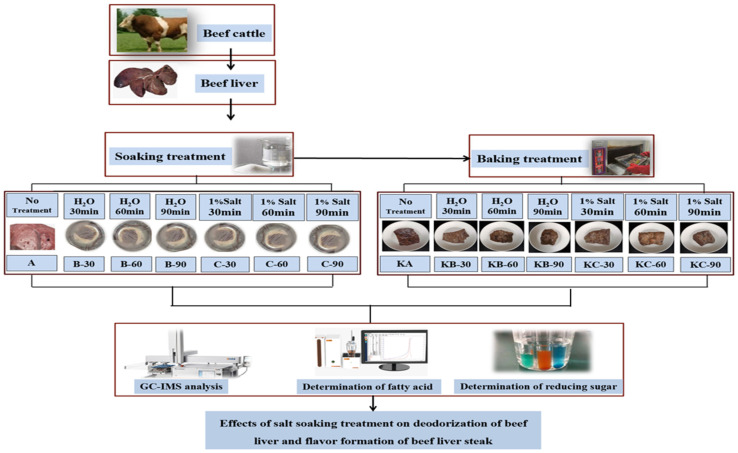
Experimental design for the effect of salt soaking treatment on the deodorization of beef liver and the flavor formation of beef liver steak.

**Figure 2 foods-12-03877-f002:**
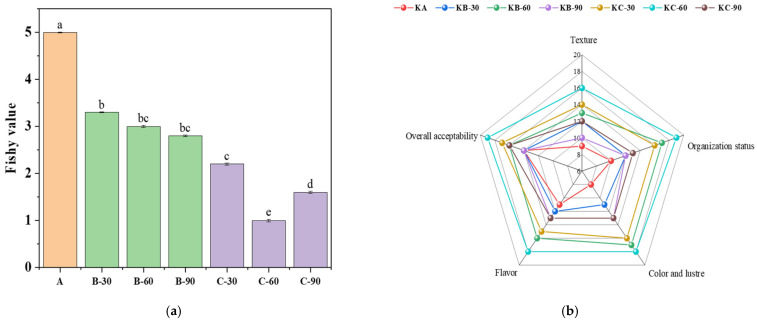
(**a**) Effect of different soaking conditions on the fishy value of beef liver. (**b**) Radar chart of the sensory evaluation score of baked beef liver steak. Note: Lower case letters in the graph represent significant differences between groups (*p* < 0.05).

**Figure 3 foods-12-03877-f003:**
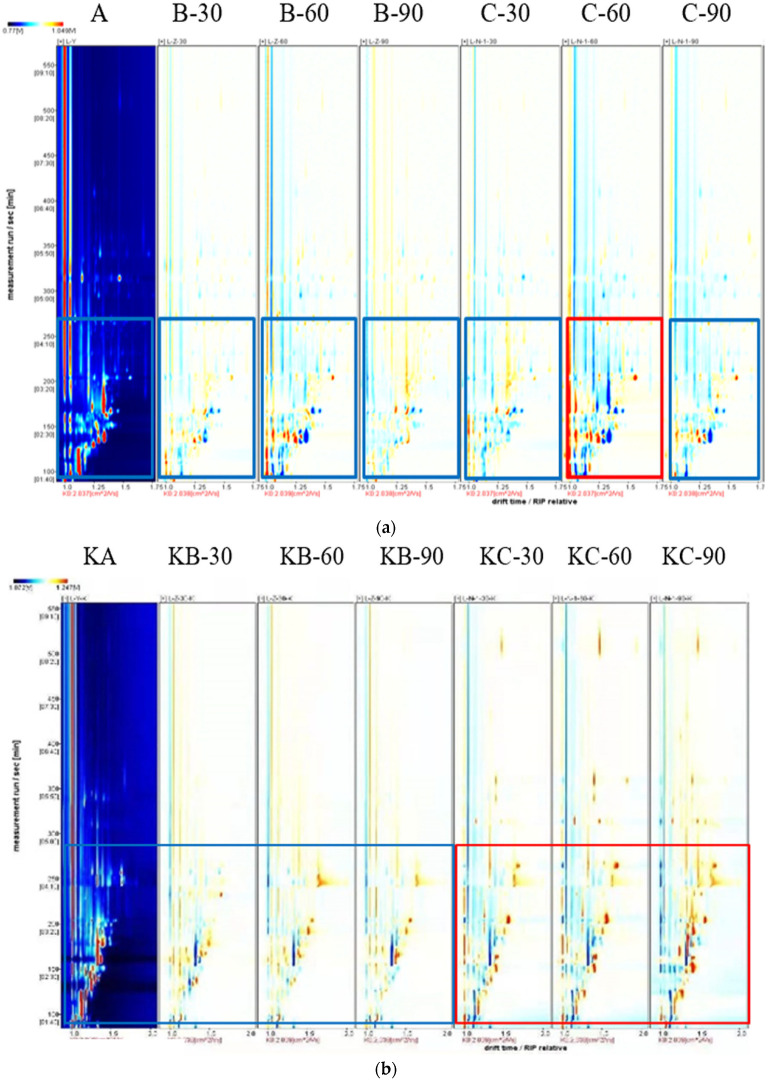
(**a**) Comparison and differential spectra of volatile components in raw beef liver under different soaking methods. (**b**) Comparison and differential spectra of volatile components in baked beef liver steak under different soaking methods. (**c**) Fingerprint patterns of beef liver and baked beef liver steak under different soaking conditions.

**Figure 4 foods-12-03877-f004:**
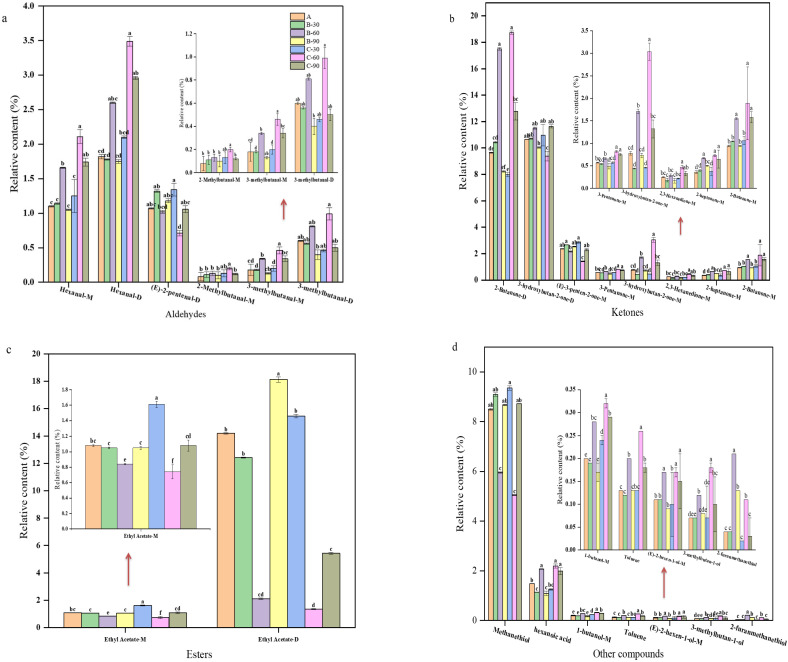
Beef liver under different soaking conditions. (**a**) Substances with significant differences in the relative content of aldehydes in beef liver between different soaking treatments. (**b**) There were significant differences in the relative content of ketones in beef liver after different soaking treatments. (**c**) Substances with significant differences in the relative content of beef liver esters between different soaking treatments. (**d**) Substances with significant differences in the relative content of other components between different treatments. Note: lowercase letters represent significant differences (*p* < 0.05).

**Figure 5 foods-12-03877-f005:**
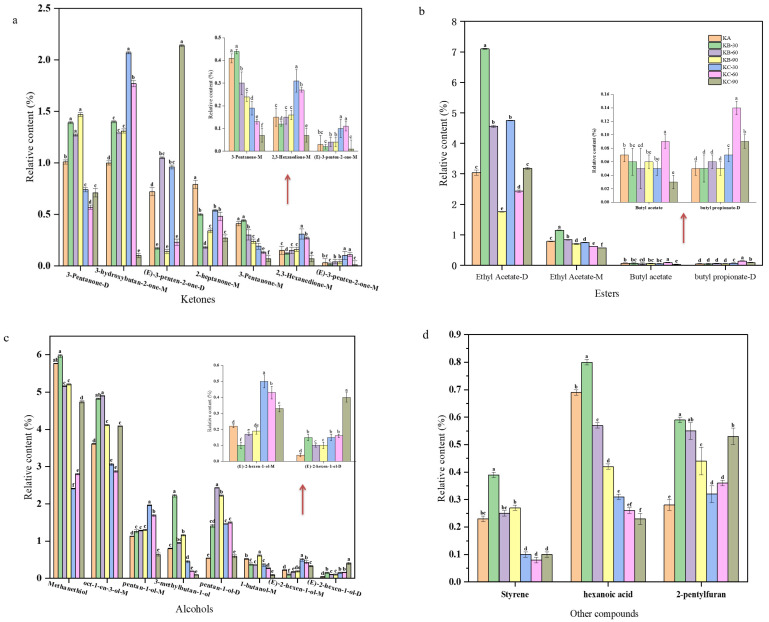
Roasted beef liver steak under different soaking conditions. (**a**) Substances with significant differences in relative ketone content. (**b**) Substances with significant differences in relative ester content. (**c**) Substances with significant differences in relative alcohol content. (**d**) Substances with significant differences in the relative content of other components. Note: lowercase letters represent significant differences (*p* < 0.05).

**Figure 6 foods-12-03877-f006:**
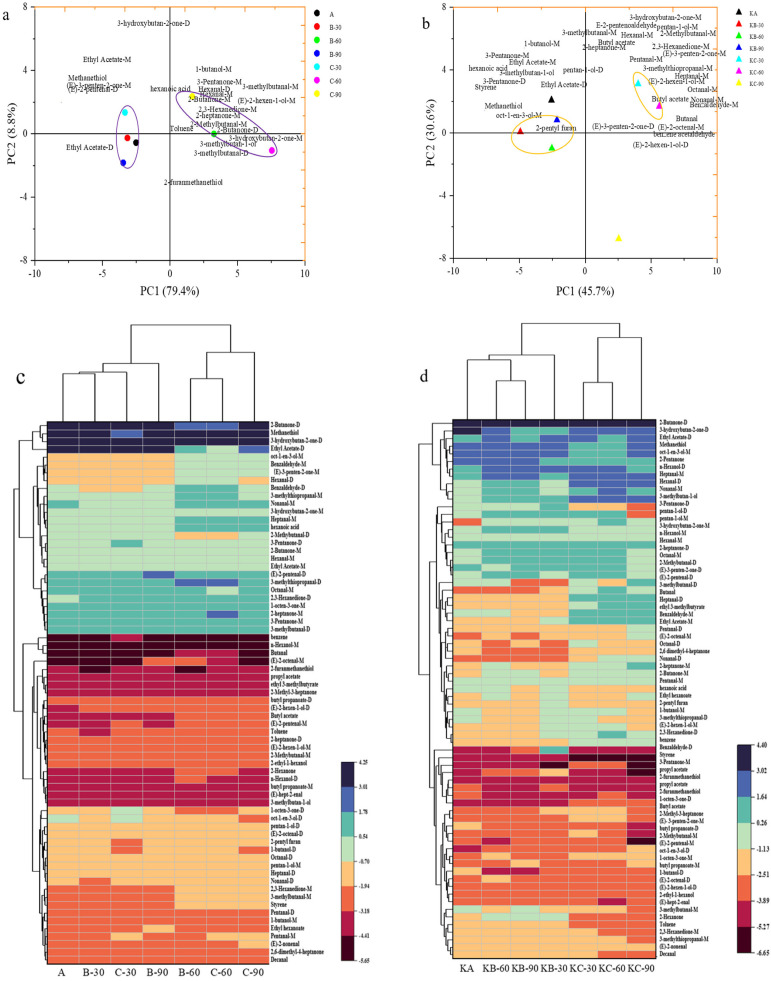
(**a**) Principal component analysis of volatile flavor compounds of beef liver under different soaking conditions. (**b**) Principal component analysis of volatile flavor compounds of beef liver steak under different soaking conditions. (**c**) Heat map of volatile components of beef liver under different soaking conditions. (**d**) Heat map of volatile components of beef liver steak under different soaking conditions.

**Figure 7 foods-12-03877-f007:**
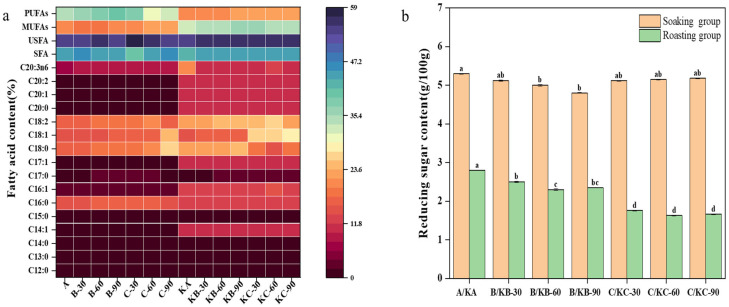
(**a**) Heatmap of fatty acid content in beef liver and beef liver steak under different soaking conditions. (**b**) Content of reducing sugar in raw beef liver and roast beef liver steak under different soaking conditions. Note: lowercase letters represent significant differences (*p* < 0.05).

**Table 1 foods-12-03877-t001:** Sample information and preparation method for salt soaking treatment.

IDX	Solution	Time	Soaking	SN	IDX	Solution	Time	Baking	SN
1	Not soaked	0	4 °C	A	1	Not soaked	0	200 °C, 7 min	KA
2	Distilled water	30	4 °C	B–30	2	Distilled water	30	200 °C, 7 min	KB–30
3	Distilled water	60	4 °C	B–60	3	Distilled water	60	200 °C, 7 min	KB–60
4	Distilled water	90	4 °C	B–90	4	Distilled water	90	200 °C, 7 min	KB–90
5	1% salt	30	4 °C	C–30	5	1% salt	30	200 °C, 7 min	KC–30
6	1% salt	60	4 °C	C–60	6	1% salt	60	200 °C, 7 min	KC–60
7	1% salt	90	4 °C	C–90	7	1% salt	90	200 °C, 7 min	KC–90

**Table 2 foods-12-03877-t002:** Compounds corresponding to some characteristic peaks of beef liver using different soaking methods.

Type	Compound	CAS#	Formula	MW	RI(s)	Dt	A	B–30	B–60	B–90	C–30	C–60	C–90
Aldehyde	Decanal	C112312	C10H20O	156.3	1277.1	1.53797	0.21 ± 0.00 ^a^	0.19 ± 0.02 ^a^	0.22 ± 0.00 ^a^	0.22 ± 0.00 ^a^	0.24 ± 0.02 ^a^	0.25 ± 0.03 ^a^	0.22 ± 0.02 ^a^
	(E)–2–nonenal	C18829566	C9H16O	140.2	1188.1	1.4107	0.25 ± 0.00 ^a^	0.22 ± 0.01 ^a^	0.26 ± 0.00 ^a^	0.22 ± 0.00 ^a^	0.25 ± 0.03 ^a^	0.24 ± 0.06 ^a^	0.27 ± 0.02 ^a^
	Nonanal–M	C124196	C9H18O	142.2	1109.1	1.47344	2.00 ± 0.01 ^a^	2.04 ± 0.07 ^a^	2.19 ± 0.02 ^a^	1.75 ± 0.01 ^a^	2.19 ± 0.02 ^a^	2.22 ± 0.02 ^a^	2.30 ± 0.07 ^a^
	Nonanal–D	C124196	C9H18O	142.2	1108.1	1.94666	0.37 ± 0.00 ^a^	0.26 ± 0.02 ^b^	0.35 ± 0.00 ^ab^	0.31 ± 0.00 ^ab^	0.35 ± 0.06 ^ab^	0.36 ± 0.04 ^ab^	0.38 ± 0.07 ^ab^
	(E)–2–octenal–M	C2548870	C8H14O	126.2	1054.7	1.33541	0.35 ± 0.00 ^b^	0.40 ± 0.01 ^ab^	0.50 ± 0.01 ^a^	0.42 ± 0.01 ^ab^	0.39 ± 0.00 ^ab^	0.43 ± 0.07 ^b^	0.45 ± 0.06 ^ab^
	(E)–2–octenal–D	C2548870	C8H14O	126.2	1054.7	1.82835	0.09 ± 0.00 ^ab^	0.07 ± 0.01 ^b^	0.10 ± 0.00 ^ab^	0.10 ± 0.00 ^a^	0.08 ± 0.01 ^ab^	0.10 ± 0.01 ^ab^	0.10 ± 0.00 ^a^
	Octanal–M	C124130	C8H16O	128.2	1008.6	1.40712	0.84 ± 0.01 ^a^	0.80 ± 0.03 ^a^	0.95 ± 0.01 ^a^	0.78 ± 0.00 ^a^	0.87 ± 0.01 ^a^	0.94 ± 0.09 ^a^	0.99 ± 0.03 ^a^
	Octanal–D	C124130	C8H16O	128.2	1004.5	1.82656	0.38 ± 0.00 ^ab^	0.28 ± 0.03 ^b^	0.44 ± 0.01 ^a^	0.33 ± 0.00 ^ab^	0.36 ± 0.02 ^ab^	0.44 ± 0.06 ^ab^	0.40 ± 0.08 ^ab^
	Benzaldehyde–M	C100527	C7H6O	106.1	956.7	1.15172	3.19 ± 0.01 ^ab^	3.16 ± 0.01 ^ab^	3.81 ± 0.02 ^a^	3.00 ± 0.03 ^bc^	2.84 ± 0.00 ^c^	3.84 ± 0.00 ^ab^	3.28 ± 0.01 ^bc^
	Benzaldehyde–D	C100527	C7H6O	106.1	957.1	1.47171	2.61 ± 0.01 ^a^	2.69 ± 0.01 ^a^	2.89 ± 0.03 ^a^	2.24 ± 0.03 ^ab^	1.96 ± 0.00 ^b^	2.36 ± 0.08 ^b^	1.91 ± 0.03 ^b^
	Benzene acetaldehyde	C122781	C8H8O	120.2	1039.3	1.25813	0.74 ± 0.00 ^a^	0.59 ± 0.03 ^ab^	0.34 ± 0.01 ^cd^	0.36 ± 0.01 ^cd^	0.64 ± 0.06 ^ab^	0.38 ± 0.00 ^cd^	0.23 ± 0.02 ^d^
	(E)–hept–2–enal	C18829555	C7H12O	112.2	952.3	1.67411	0.14 ± 0.00 ^ab^	0.13 ± 0.00 ^ab^	0.17 ± 0.00 ^a^	0.13 ± 0.00 ^ab^	0.12 ± 0.09 ^b^	0.15 ± 0.04 ^ab^	0.13 ± 0.01 ^b^
	Hexanal–M	C66251	C6H12O	100.2	791.9	1.25392	1.10 ± 0.01 ^c^	1.14 ± 0.01 ^c^	1.66 ± 0.01 ^b^	1.05 ± 0.01 ^c^	1.25 ± 0.24 ^c^	2.11 ± 0.10 ^a^	1.74 ± 0.06 ^ab^
	Hexanal–D	C66251	C6H12O	100.2	787.2	1.56558	1.82 ± 0.03 ^cd^	1.78 ± 0.01 ^cd^	2.60 ± 0.01 ^abc^	1.75 ± 0.03 ^cd^	2.10 ± 0.02 ^bcd^	3.49 ± 0.07 ^a^	2.96 ± 0.02 ^ab^
	Heptanal–M	C111717	C7H14O	114.2	898.2	1.32579	1.35 ± 0.02 ^c^	1.65 ± 0.01 ^abc^	1.79 ± 0.01 ^abc^	1.53 ± 0.01 ^bc^	1.97 ± 0.02 ^a^	1.88 ± 0.01 ^abc^	1.92 ± 0.07 ^ab^
	Heptanal–D	C111717	C7H14O	114.2	898.9	1.69956	0.27 ± 0.00 ^ab^	0.31 ± 0.00 ^ab^	0.35 ± 0.00 ^ab^	0.27 ± 0.00 ^ab^	0.33 ± 0.09 ^ab^	0.37 ± 0.04 ^ab^	0.41 ± 0.08 ^a^
	3–methylthiopropanal–M	C3268493	C4H8OS	104.2	905.2	1.08897	1.93 ± 0.04 ^bcd^	2.19 ± 0.01 ^abc^	2.73 ± 0.03 ^a^	1.56 ± 0.02 ^d^	1.61 ± 0.06 ^d^	2.76 ± 0.07 ^ab^	2.16 ± 0.09 ^abcd^
	3–methylthiopropanal–D	C3268493	C4H8OS	104.2	904.7	1.40189	0.86 ± 0.03 ^abc^	1.15 ± 0.02 ^ab^	1.32 ± 0.04 ^a^	0.65 ± 0.02 ^c^	0.64 ± 0.06 ^c^	1.13 ± 0.08 ^abc^	0.83 ± 0.05 ^bc^
	Pentanal–M	C110623	C5H10O	86.1	693.4	1.183 21	0.25 ± 0.00 ^a^	0.24 ± 0.00 ^ab^	0.22 ± 0.00 ^c^	0.22 ± 0.00 ^bc^	0.27 ± 0.02 ^a^	0.32 ± 0.05 ^a^	0.28 ± 0.06 ^a^
	Pentanal–D	C110623	C5H10O	86.1	692.1	1.43008	0.26 ± 0.00 ^a^	0.15 ± 0.00 ^cd^	0.16 ± 0.00 ^cd^	0.22 ± 0.00 ^ab^	0.20 ± 0.01 ^bc^	0.17 ± 0.02 ^cd^	0.19 ± 0.00 ^bcd^
	Butanal	C123728	C4H8O	72.1	540.3	1.28464	0.44 ± 0.01 ^b^	0.57 ± 0.00 ^a^	0.48 ± 0.00 ^b^	0.36 ± 0.00 ^c^	0.47 ± 0.02 ^b^	0.39 ± 0.01 ^c^	0.36 ± 0.01 ^c^
	(E)–2–pentenal–M	C1576870	C5H8O	84.1	744.8	1.10603	0.03 ± 0.00 ^cd^	0.02 ± 0.00 ^d^	0.05 ± 0.00 ^b^	0.03 ± 0.00 ^cd^	0.02 ± 0.00 ^d^	0.08 ± 0.00 ^a^	0.04 ± 0.00 ^bc^
	(E)–2–pentenal–D	C1576870	C5H8O	84.1	745.3	1.35813	1.07 ± 0.01 ^abc^	1.32 ± 0.00 ^a^	1.02 ± 0.02 ^bcd^	1.18 ± 0.03 ^abc^	1.34 ± 0.09 ^ab^	0.71 ± 0.04 ^d^	1.06 ± 0.05 ^abc^
	2–methylbutanal–M	C96173	C5H10O	86.1	652.8	1.15776	0.08 ± 0.00 ^b^	0.11 ± 0.00 ^b^	0.13 ± 0.00 ^b^	0.10 ± 0.00 ^b^	0.13 ± 0.05 ^b^	0.20 ± 0.02 ^a^	0.12 ± 0.00 ^b^
	2–methylbutanal–D	C96173	C5H10O	86.1	654.4	1.40175	1.36 ± 0.03 ^a^	1.38 ± 0.01 ^a^	1.96 ± 0.03 ^a^	0.95 ± 0.01 ^a^	0.92 ± 0.07 ^b^	2.09 ± 0.01 ^a^	1.22 ± 0.07 ^a^
	3–methylbutanal–M	C590863	C5H10O	86.1	635.1	1.17318	0.18 ± 0.00 ^cd^	0.18 ± 0.00 ^d^	0.34 ± 0.00 ^b^	0.13 ± 0.00 ^cd^	0.20 ± 0.04 ^d^	0.46 ± 0.05 ^a^	0.34 ± 0.04 ^bc^
	3–methylbutanal–D	C590863	C5H10O	86.1	638.4	1.41016	0.60 ± 0.01 ^ab^	0.56 ± 0.01 ^ab^	0.81 ± 0.01 ^ab^	0.40 ± 0.00 ^b^	0.46 ± 0.02 ^ab^	0.99 ± 0.09 ^c^	0.50 ± 0.05 ^ab^
Esters	Butyl propanoate–M	C590012	C7H14O2	130.2	907.3	1.28334	0.15 ± 0.00 ^a^	0.14 ± 0.01 ^ab^	0.14 ± 0.00 ^ab^	0.13 ± 0.00 ^ab^	0.14 ± 0.01 ^ab^	0.14 ± 0.01 ^b^	0.12 ± 0.01 ^b^
	Butyl propanoate–D	C590012	C7H14O2	130.2	907.1	1.72743	0.06 ± 0.00 ^a^	0.05 ± 0.00 ^a^	0.06 ± 0.00 ^a^	0.05 ± 0.00 ^a^	0.06 ± 0.00 ^a^	0.06 ± 0.02 ^a^	0.06 ± 0.05 ^a^
	Ethyl 3–methylbutyrate	C108645	C7H14O2	130.2	842.7	1.65769	0.07 ± 0.00 ^ab^	0.05 ± 0.00 ^b^	0.07 ± 0.00 ^b^	0.09 ± 0.00 ^a^	0.08 ± 0.00 ^ab^	0.08 ± 0.01 ^b^	0.07 ± 0.02 ^b^
	Propyl acetate	C109604	C5H10O2	102.1	703.6	1.47843	0.10 ± 0.00 ^a^	0.04 ± 0.00 ^bc^	0.04 ± 0.00 ^c^	0.10 ± 0.00 ^a^	0.07 ± 0.00 ^abc^	0.05 ± 0.00 ^bc^	0.05 ± 0.02 ^bc^
	Butyl acetate	C123864	C6H12O2	116.2	816.1	1.61605	0.03 ± 0.00 ^a^	0.02 ± 0.00 ^b^	0.04 ± 0.00 ^a^	0.03 ± 0.00 ^a^	0.03 ± 0.00 ^a^	0.04 ± 0.00 ^a^	0.04 ± 0.00 ^a^
	Ethyl acetate–M	C141786	C4H8O2	88.1	599.1	1.0987	1.08 ± 0.01 ^bc^	1.05 ± 0.00 ^c^	0.84 ± 0.01 ^e^	1.05 ± 0.00 ^c^	1.61 ± 0.04 ^a^	0.74 ± 0.09 ^f^	1.08 ± 0.07 ^cd^
	Ethyl acetate–D	C141786	C4H8O2	88.1	601.1	1.34402	14.20 ± 0.05 ^b^	12.43 ± 0.04 ^b^	2.10 ± 0.06 ^cd^	18.12 ± 0.19 ^a^	15.45 ± 0.14 ^b^	1.35 ± 0.03 ^d^	5.43 ± 0.07 ^c^
	Ethyl hexanoate	C123660	C8H16O2	144.2	999.5	1.34258	0.20 ± 0.00 ^b^	0.24 ± 0.01 ^ab^	0.23 ± 0.00 ^b^	0.28 ± 0.00 ^a^	0.23 ± 0.05 ^b^	0.23 ± 0.06 ^b^	0.26 ± 0.01 ^ab^
Ketones	2–methyl–3–heptanone	C13019200	C8H16O	128.2	1085.8	1.27626	0.06 ± 0.00 ^ab^	0.05 ± 0.00 ^b^	0.06 ± 0.00 ^ab^	0.07 ± 0.00 ^ab^	0.06 ± 0.01 ^ab^	0.06 ± 0.01 ^b^	0.06 ± 0.01 ^b^
	2,6 dimethyl–4–heptanone	C108838	C9H18O	142.2	967.8	1.32856	0.21 ± 0.00 ^ab^	0.21 ± 0.01 ^a^	0.19 ± 0.00 ^a^	0.21 ± 0.00 ^a^	0.25 ± 0.00 ^a^	0.20 ± 0.02 ^b^	0.19 ± 0.06 ^b^
	3–pentanone–M	C96220	C5H10O	86.1	693.2	1.10836	0.57 ± 0.00 ^bc^	0.54 ± 0.01 ^cd^	0.66 ± 0.00 ^b^	0.49 ± 0.00 ^d^	0.57 ± 0.02 ^cd^	0.82 ± 0.02 ^a^	0.75 ± 0.03 ^a^
	3–pentanone–D	C96220	C5H10O	86.1	692.6	1.35946	1.01 ± 0.01 ^cd^	0.95 ± 0.01 ^d^	1.43 ± 0.00 ^a^	1.23 ± 0.01 ^ab^	1.05 ± 0.02 ^cd^	1.37 ± 0.03 ^abc^	1.41 ± 0.03 ^a^
	2–hexanone	C591786	C6H12O	100.2	779.5	1.18752	0.14 ± 0.00 ^a^	0.14 ± 0.00 ^a^	0.16 ± 0.00 ^a^	0.15 ± 0.00 ^a^	0.12 ± 0.07 ^b^	0.19 ± 0.02 ^a^	0.16 ± 0.01 ^a^
	3–hydroxybutan–2–one–M	C513860	C4H8O2	88.1	712.8	1.05627	0.78 ± 0.04 ^cd^	0.44 ± 0.01 ^d^	1.71 ± 0.05 ^b^	0.73 ± 0.05 ^cd^	0.46 ± 0.01 ^d^	3.04 ± 0.19 ^a^	1.32 ± 0.20 ^bc^
	3–hydroxybutan–2–one–D	C513860	C4H8O2	88.1	713.2	1.33282	10.65 ± 0.01 ^ab^	10.71 ± 0.04 ^ab^	11.50 ± 0.08 ^ab^	10.04 ± 0.06 ^b^	10.97 ± 0.81 ^ab^	9.39 ± 0.37 ^c^	11.64 ± 0.12 ^ab^
	2,3–hexanedione–M	C3848246	C6H10O2	114.1	778.3	1.09198	0.25 ± 0.01 ^bc^	0.18 ± 0.00 ^c^	0.29 ± 0.00 ^bc^	0.17 ± 0.00 ^c^	0.22 ± 0.01 ^bc^	0.47 ± 0.04 ^a^	0.33 ± 0.05 ^ab^
	2,3–hexanedione–D	C3848246	C6H10O2	114.1	772.7	1.35966	0.73 ± 0.01 ^abc^	0.86 ± 0.03 ^ab^	0.57 ± 0.02 ^bc^	0.66 ± 0.02 ^bc^	1.09 ± 0.08 ^a^	0.55 ± 0.00 ^c^	0.70 ± 0.01 ^bc^
	(E)–3–penten–2–one–M	C3102338	C5H8O	84.1	731.8	1.35105	2.37 ± 0.03 ^ab^	2.67 ± 0.01 ^a^	2.17 ± 0.04 ^b^	2.54 ± 0.04 ^ab^	2.87 ± 0.08 ^a^	1.42 ± 0.04 ^c^	2.30 ± 0.08 ^ab^
	2–heptanone–M	C110430	C7H14O	114.2	891.8	1.25627	0.36 ± 0.00 ^d^	0.40 ± 0.00 ^d^	0.67 ± 0.00 ^a^	0.50 ± 0.01 ^bc^	0.38 ± 0.09 ^d^	0.73 ± 0.02 ^a^	0.65 ± 0.20 ^a^
	2–heptanone–D	C110430	C7H14O	114.2	890.1	1.63201	0.11 ± 0.00 ^c^	0.15 ± 0.01 ^bc^	0.23 ± 0.00 ^a^	0.20 ± 0.00 ^a^	0.13 ± 0.00 ^c^	0.21 ± 0.08 ^ab^	0.23 ± 0.08 ^a^
	1–octen–3–one–M	C4312996	C8H14O	126.2	979.4	1.27092	0.46 ± 0.00 ^d^	0.60 ± 0.00 ^b^	0.76 ± 0.00 ^a^	0.56 ± 0.00 ^bc^	0.45 ± 0.01 ^d^	0.69 ± 0.00 ^b^	0.65 ± 0.08 ^b^
	1–octen–3–one–D	C4312996	C8H14O	126.2	978.7	1.68592	0.06 ± 0.00 ^ab^	0.10 ± 0.00 ^ab^	0.12 ± 0.00 ^a^	0.10 ± 0.00 ^bc^	0.07 ± 0.03 ^c^	0.09 ± 0.10 ^a^	0.09 ± 0.02 ^bc^
	2–butanone–M	C78933	C4H8O	72.1	565.1	1.06071	0.94 ± 0.02 ^b^	1.04 ± 0.01 ^b^	1.55 ± 0.02 ^a^	0.94 ± 0.01 ^b^	1.06 ± 0.09 ^b^	1.89 ± 0.81 ^a^	1.58 ± 0.12 ^a^
	2–butanone–D	C78933	C4H8O	72.1	566.8	1.24553	9.65 ± 0.04 ^de^	10.44 ± 0.07 ^cd^	17.52 ± 0.10 ^a^	8.22 ± 0.06 ^ef^	7.99 ± 0.15 ^e^	18.74 ± 0.13 ^a^	12.78 ± 0.66 ^bc^
	2–ethyl–1–hexanol	C104767	C8H18O	130.2	1026.9	1.26013	0.15 ± 0.00 ^a^	0.14 ± 0.01 ^ab^	0.18 ± 0.00 ^ab^	0.15 ± 0.00 ^ab^	0.12 ± 0.00 ^ab^	0.16 ± 0.01 ^b^	0.17 ± 0.01 ^b^
	Oct–1–en–3–ol–M	C3391864	C8H16O	128.2	986	1.15975	3.33 ± 0.01 ^a^	3.26 ± 0.05 ^a^	2.99 ± 0.02 ^a^	3.51 ± 0.03 ^a^	2.70 ± 0.20 ^a^	3.02 ± 0.18 ^a^	2.47 ± 0.07 ^a^
	Oct–1–en–3–ol–D	C3391864	C8H16O	128.2	979.4	1.60406	0.10 ± 0.00 ^ab^	0.08 ± 0.00 ^b^	0.11 ± 0.00 ^b^	0.09 ± 0.00 ^a^	0.10 ± 0.01 ^ab^	0.11 ± 0.01 ^b^	0.10 ± 0.03 ^b^
	n–hexanol–M	C111273	C6H14O	102.2	861.9	1.32497	0.28 ± 0.01 ^a^	0.48 ± 0.01 ^bc^	0.23 ± 0.00 ^c^	0.44 ± 0.02 ^a^	0.65 ± 0.06 ^abc^	0.24 ± 0.01 ^bc^	0.30 ± 0.02 ^bc^
	n–hexanol–D	C111273	C6H14O	102.2	861.9	1.6385	0.13 ± 0.00 ^a^	0.09 ± 0.01 ^b^	0.14 ± 0.00 ^a^	0.12 ± 0.00 ^a^	0.13 ± 0.00 ^a^	0.15 ± 0.09 ^a^	0.12 ± 0.01 ^a^
	(E)–2–hexen–1–ol–M	C928950	C6H12O	100.2	846	1.18518	0.11 ± 0.00 ^b^	0.11 ± 0.00 ^b^	0.17 ± 0.00 ^a^	0.09 ± 0.00 ^b^	0.10 ± 0.07 ^b^	0.17 ± 0.01 ^a^	0.15 ± 0.06 ^a^
	(E)–2–hexen–1–ol–D	C928950	C6H12O	100.2	844	1.51752	0.03 ± 0.00 ^b^	0.03 ± 0.00 ^b^	0.02 ± 0.00 ^cd^	0.03 ± 0.00 ^a^	0.05 ± 0.03 ^b^	0.02 ± 0.05 ^d^	0.02 ± 0.01 ^c^
	3–methylbutan–1–ol	C123513	C5H12O	88.1	727.3	1.48806	0.07 ± 0.00 ^b^	0.07 ± 0.00 ^ab^	0.12 ± 0.00 ^b^	0.08 ± 0.00 ^a^	0.07 ± 0.01 ^b^	0.18 ± 0.01 ^b^	0.10 ± 0.01 ^ab^
	1–butanol–M	C71363	C4H10O	74.1	654.3	1.1823	0.20 ± 0.00 ^ab^	0.19 ± 0.00 ^b^	0.28 ± 0.00 ^ab^	0.17 ± 0.00 ^ab^	0.24 ± 0.02 ^ab^	0.32 ± 0.01 ^b^	0.29 ± 0.01 ^b^
	1–butanol–D	C71363	C4H10O	74.1	653	1.37683	0.32 ± 0.00 ^ab^	0.32 ± 0.00 ^ab^	0.42 ± 0.01 ^b^	0.34 ± 0.00 ^ab^	0.23 ± 0.01 ^a^	0.39 ± 0.03 ^b^	0.20 ± 0.03 ^b^
	Methanethiol	C74931	CH4S	48.1	441.9	1.0483	8.50 ± 0.01 ^bc^	9.09 ± 0.04 ^cd^	5.96 ± 0.07 ^b^	8.67 ± 0.03 ^d^	9.35 ± 0.07 ^cd^	5.05 ± 0.09 ^a^	8.72 ± 0.02 ^a^
	2–furanmethanethiol	C98022	C5H6OS	114.2	912.8	1.11099	0.04 ± 0.00 ^cd^	0.04 ± 0.00 ^d^	0.21 ± 0.00 ^a^	0.13 ± 0.00 ^ab^	0.02 ± 0.00 ^cd^	0.11 ± 0.00 ^abc^	0.03 ± 0.04 ^a^
	Pentan–1–ol–M	C71410	C5H12O	88.1	762.4	1.25392	0.29 ± 0.00 ^a^	0.34 ± 0.01 ^a^	0.34 ± 0.00 ^a^	0.30 ± 0.01 ^a^	0.38 ± 0.09 ^a^	0.34 ± 0.04 ^a^	0.31 ± 0.01 ^a^
	Pentan–1–ol–D	C71410	C5H12O	88.1	764.4	1.51286	0.09 ± 0.00 ^ab^	0.09 ± 0.00 ^ab^	0.11 ± 0.00 ^a^	0.09 ± 0.00 ^ab^	0.08 ± 0.00 ^b^	0.10 ± 0.01 ^ab^	0.09 ± 0.01 ^ab^
Hydrocarbon	Styrene	C100425	C8H8	104.2	887.9	1.41987	0.19 ± 0.00 ^abc^	0.22 ± 0.00 ^a^	0.19 ± 0.00 ^bcd^	0.21 ± 0.00 ^ab^	0.17 ± 0.01 ^cd^	0.15 ± 0.01 ^d^	0.17 ± 0.01 ^cd^
	Toluene	C108883	C7H8	92.1	760.5	1.01492	0.13 ± 0.00 ^cde^	0.12 ± 0.00 ^e^	0.20 ± 0.00 ^b^	0.13 ± 0.00 ^cde^	0.13 ± 0.00 ^de^	0.26 ± 0.00 ^a^	0.18 ± 0.01 ^bc^
Acids	Hexanoic acid	C142621	C6H12O2	116.2	994.1	1.30503	1.48 ± 0.01 c	1.14 ± 0.00 ^e^	2.07 ± 0.01 ^b^	1.09 ± 0.01 ^e^	1.26 ± 0.03 ^de^	2.19 ± 0.08 ^b^	1.99 ± 0.14 ^b^
Furan	2–pentyl furan	C3777693	C9H14O	138.2	994.1	1.2552	0.31 ± 0.00 ^a^	0.30 ± 0.01 ^a^	0.35 ± 0.00 ^a^	0.31 ± 0.00 ^a^	0.21 ± 0.00 ^c^	0.35 ± 0.02 ^a^	0.32 ± 0.19 ^a^

Note: Different lowercase letters represent significant differences between different soaking treatments (*p* < 0.05). Abbreviations: DT, drift time; MW, molecular weight; RI, retention index.

**Table 3 foods-12-03877-t003:** Compounds corresponding to partial characteristic peaks of beef liver steak under different soaking conditions.

Type	Compound	CAS#	Formula	MW	RI(s)	Dt	KA	KB–30	KB–60	KB–90	KC–30	KC–60	KC–90
Aldehyde	Decanal	C112312	C10H20O	156.3	1277.1	1.53797	0.20 ± 0.00 ^bc^	0.26 ± 0.02 ^a^	0.21 ± 0.00 ^ab^	0.25 ± 0.00 ^a^	0.18 ± 0.02 ^bc^	0.15 ± 0.02 ^c^	0.17 ± 0.02 ^bc^
	(E)–2–nonenal	C18829566	C9H16O	140.2	1188.1	1.4107	0.21 ± 0.00 ^abc^	0.23 ± 0.01 ^ab^	0.19 ± 0.00 ^bc^	0.20 ± 0.00 ^bc^	0.19 ± 0.07 ^c^	0.18 ± 0.01 ^c^	0.23 ± 0.01 ^a^
	Nonanal–M	C124196	C9H18O	142.2	1109.1	1.47344	1.19 ± 0.04 ^g^	1.26 ± 0.07 ^g^	1.59 ± 0.02 ^f^	1.84 ± 0.01 ^e^	3.73 ± 0.22 ^c^	5.48 ± 0.07 ^a^	3.99 ± 0.07 ^b^
	Nonanal–D	C124196	C9H18O	142.2	1108.1	1.94666	0.33 ± 0.01 ^de^	0.38 ± 0.02 ^d^	0.29 ± 0.00 ^e^	0.33 ± 0.00 ^de^	0.47 ± 0.06 ^c^	1.03 ± 0.02 ^a^	0.56 ± 0.02 ^b^
	(E)–2–octenal–M	C2548870	C8H14O	126.2	1054.7	1.33541	0.46 ± 0.01 ^f^	0.52 ± 0.01 ^e^	0.76 ± 0.01 ^c^	0.69 ± 0.01 ^d^	0.88 ± 0.04 ^b^	0.77 ± 0.01 ^c^	1.13 ± 0.01 ^a^
	(E)–2–octenal–D	C2548870	C8H14O	126.2	1054.7	1.82835	0.09 ± 0.00 ^ab^	0.10 ± 0.01 ^a^	0.09 ± 0.00 ^ab^	0.08 ± 0.00 ^abc^	0.07 ± 0.14 ^bc^	0.06 ± 0.01 ^c^	0.10 ± 0.01 ^ab^
	Octanal–M	C124130	C8H16O	128.2	1008.6	1.40712	0.83 ± 0.01 ^f^	1.13 ± 0.03 ^e^	1.54 ± 0.01 ^c^	1.45 ± 0.00 ^cd^	2.22 ± 0.11 ^b^	3.43 ± 0.03 ^a^	2.29 ± 0.03 ^b^
	Octanal–D	C124130	C8H16O	128.2	1004.5	1.82656	0.42 ± 0.00 ^de^	0.46 ± 0.03 ^cd^	0.33 ± 0.01 ^e^	0.36 ± 0.00 ^e^	0.50 ± 0.10 ^c^	1.23 ± 0.03 ^a^	0.58 ± 0. 03 ^b^
	Benzaldehyde–M	C100527	C7H6O	106.1	956.7	1.15172	0.32 ± 0.01 ^ef^	0.55 ± 0.01 ^d^	0.43 ± 0.02 ^e^	0.36 ± 0.03 ^ef^	1.29 ± 0.10 ^c^	1.81 ± 0.01 ^a^	1.41 ± 0.01 ^b^
	Benzaldehyde–D	C100527	C7H6O	106.1	957.1	1.47171	0.13 ± 0.01 ^d^	0.11 ± 0.01 ^d^	0.08 ± 0.03 ^d^	0.11 ± 0.03 ^d^	0.36 ± 0.10 ^c^	0.58 ± 0.01 ^a^	0.72 ± 0.01 ^b^
	Benzene acetaldehyde	C122781	C8H8O	120.2	1039.3	1.25813	0.17 ± 0.02 ^e^	0.19 ± 0.03 ^e^	0.18 ± 0.01 ^e^	0.16 ± 0.01 ^e^	0.26 ± 0.06 ^d^	0.34 ± 0.03 ^c^	0.58 ± 0.03 ^a^
	(E)–hept–2–enal	C18829555	C7H12O	112.2	952.3	1.67411	0.09 ± 0.00 ^d^	0.11 ± 0.00 ^cd^	0.16 ± 0.00 ^b^	0.19 ± 0.00 ^b^	0.15 ± 0.01 ^bc^	0.17 ± 0.00 ^b^	0.34 ± 0.00 ^a^
	Hexanal–M	C66251	C6H12O	100.2	791.9	1.25392	1.18 ± 0.02 ^d^	1.00 ± 0.01 ^e^	1.36 ± 0.01 ^b^	1.41 ± 0.01 ^b^	1.74 ± 0.24 ^a^	1.33 ± 0.01 ^bc^	0.77 ± 0.01 ^f^
	Hexanal–D	C66251	C6H12O	100.2	787.2	1.56558	0.52 ± 0.01 ^f^	1.08 ± 0.01 ^e^	1.87 ± 0.01 ^d^	2.59 ± 0.03 ^c^	6.58 ± 0.02 ^a^	6.90 ± 0.00 ^a^	3.71 ± 0.01 ^b^
	Heptanal–M	C111717	C7H14O	114.2	898.2	1.32579	2.45 ± 0.02 ^f^	2.44 ± 0.01 ^f^	3.60 ± 0.01 ^cde^	3.73 ± 0.01 ^e^	5.35 ± 0.12 ^a^	4.47 ± 0.01 ^b^	3.64 ± 0.01 ^cd^
	Heptanal–D	C111717	C7H14O	114.2	898.9	1.69956	0.08 ± 0.01 ^d^	0.19 ± 0.00 ^d^	0.16 ± 0.00 ^d^	0.16 ± 0.00 ^d^	0.84 ± 0.09 ^c^	1.92 ± 0.00 ^a^	1.02 ± 0.01 ^b^
	3–methylthiopropanal–M	C3268493	C4H8OS	104.2	905.2	1.08897	0.19 ± 0.02 ^c^	0.14 ± 0.01 ^d^	0.10 ± 0.03 ^g^	0.13 ± 0.02 ^e^	0.26 ± 0.06 ^b^	0.33 ± 0.01 ^a^	0.19 ± 0.01 ^c^
	3–methylthiopropanal–D	C3268493	C4H8OS	104.2	904.7	1.40189	0.35 ± 0.02 ^c^	0.51 ± 0.02 ^b^	0.66 ± 0.04 ^a^	0.73 ± 0.02 ^a^	0.13 ± 0.01 ^f^	0.16 ± 0.01 ^ef^	0.11 ± 0.01 ^f^
	Pentanal–M	C110623	C5H10O	86.1	693.4	1.183 21	0.30 ± 0.00 ^d^	0.50 ± 0.00 ^c^	0.26 ± 0.00 ^de^	0.23 ± 0.00 ^e^	0.73 ± 0.02 ^a^	0.65 ± 0.01 ^b^	0.28 ± 0.01 ^de^
	Pentanal–D	C110623	C5H10O	86.1	692.1	1.43008	0.16 ± 0.01 ^g^	0.65 ± 0.01 ^de^	0.76 ± 0.01 ^d^	1.08 ± 0.00 ^bc^	1.16 ± 0.05 ^b^	1.75 ± 0.01 ^a^	0.84 ± 0.01 ^cd^
	Butanal	C123728	C4H8O	72.1	540.3	1.28464	0.22 ± 0.01 ^e^	0.19 ± 0.00 ^e^	0.23 ± 0.00 ^e^	0.29 ± 0.00 ^de^	0.70 ± 0.09 ^c^	1.44 ± 0.01 ^b^	1.66 ± 0.01 ^a^
	(E)–2–pentenal–M	C1576870	C5H8O	84.1	744.8	1.10603	0.13 ± 0.00 ^a^	0.08 ± 0.00 ^c^	0.06 ± 0.00 ^d^	0.10 ± 0.00 ^b^	0.13 ± 0.02 ^a^	0.11 ± 0.01 ^b^	0.02 ± 0.00 ^e^
	(E)–2–pentenal–D	C1576870	C5H8O	84.1	745.3	1.35813	0.40 ± 0.01 ^d^	0.28 ± 0.00 ^d^	0.61 ± 0.02 ^c^	0.31 ± 0.01 ^d^	0.63 ± 0.01 ^c^	0.71 ± 0.00 ^bc^	2.02 ± 0.01 ^a^
	2–methylbutanal–M	C96173	C5H10O	86.1	652.8	1.15776	0.15 ± 0.00 ^c^	0.07 ± 0.00 ^f^	0.07 ± 0.00 ^f^	0.10 ± 0.00 ^e^	0.24 ± 0.01 ^a^	0.16 ± 0.00 ^b^	0.04 ± 0.00 ^g^
	2–methylbutanal–D	C96173	C5H10O	86.1	654.4	1.40175	0.72 ± 0.03 ^e^	0.45 ± 0.01 ^f^	0.46 ± 0.03 ^f^	0.40 ± 0.01 ^f^	1.95 ± 0.07 ^d^	2.14 ± 0.01 ^c^	3.02 ± 0.01 ^a^
	3–methylbutanal–M	C590863	C5H10O	86.1	635.1	1.17318	0.39 ± 0.00 ^a^	0.24 ± 0.00 ^e^	0.22 ± 0.00 ^e^	0.28 ± 0.00 ^cd^	0.30 ± 0.01 ^bc^	0.25 ± 0.00 ^de^	0.13 ± 0.01 ^f^
	3–methylbutanal–D	C590863	C5H10O	86.1	638.4	1.41016	0.43 ± 0.01 ^d^	0.28 ± 0.00 ^e^	0.21 ± 0.01 ^e^	0.19 ± 0.01 ^e^	1.33 ± 0.02 ^c^	1.60 ± 0.00 ^b^	2.30 ± 0.00 ^a^
Esters	Butyl propanoate–M	C590012	C7H14O2	130.2	907.3	1.28334	0.08 ± 0.00 ^c^	0.17 ± 0.00 ^a^	0.11 ± 0.00 ^b^	0.09 ± 0.00 ^bc^	0.09 ± 0.00 ^bc^	0.10 ± 0.01 ^bc^	0.08 ± 0.01 ^c^
	Butyl propanoate–D	C590012	C7H14O2	130.2	907.1	1.72743	0.05 ± 0.00 ^d^	0.05 ± 0.00 ^d^	0.06 ± 0.00 ^d^	0.05 ± 0.00 ^d^	0.07 ± 0.00 ^c^	0.14 ± 0.00 ^a^	0.09 ± 0.00 ^b^
	Ethyl 3–methylbutyrate	C108645	C7H14O2	130.2	842.7	1.65769	0.05 ± 0.00 ^b^	2.03 ± 0.00 ^a^	0.05 ± 0.00 ^b^	0.08 ± 0.00 ^b^	0.04 ± 0.00 ^b^	0.04 ± 0.00 ^b^	0.04 ± 0.00 ^b^
	Propyl acetate	C109604	C5H10O2	102.1	703.6	1.47843	0.05 ± 0.00 ^cd^	0.22 ± 0.01 ^a^	0.07 ± 0.00 ^c^	0.11 ± 0.00 ^b^	0.03 ± 0.00 ^d^	0.03 ± 0.00 ^d^	0.02 ± 0.00 ^d^
	Butyl acetate	C123864	C6H12O2	116.2	816.1	1.61605	0.07 ± 0.00 ^b^	0.06 ± 0.00 ^bc^	0.05 ± 0.00 ^cd^	0.06 ± 0.00 ^bc^	0.05 ± 0.00 ^bc^	0.09 ± 0.00 ^a^	0.03 ± 0.00 ^e^
	Ethyl acetate–M	C141786	C4H8O2	88.1	599.1	1.0987	0.79 ± 0.01 ^c^	1.15 ± 0.00 ^a^	0.84 ± 0.01 ^b^	0.70 ± 0.00 ^d^	0.75 ± 0.01 ^d^	0.62 ± 0.01 ^e^	0.57 ± 0.01 ^f^
	Ethyl acetate–D	C141786	C4H8O2	88.1	601.1	1.34402	3.04 ± 0.10 ^c^	7.10 ± 0.04 ^a^	4.56 ± 0.02 ^b^	1.76 ± 0.09 ^e^	4.75 ± 0.01 ^b^	2.43 ± 0.04 ^d^	3.18 ± 0.04 ^c^
	Ethyl hexanoate	C123660	C8H16O2	144.2	999.5	1.34258	0.28 ± 0.00 ^d^	0.59 ± 0.01 ^a^	0.26 ± 0.00 ^d^	0.27 ± 0.00 ^d^	0.33 ± 0.06 ^bc^	0.29 ± 0.01 ^cd^	0.35 ± 0.01 ^b^
Ketones	2–methyl–3–heptanone	C13019200	C8H16O	128.2	1085.8	1.27626	0.07 ± 0.00 ^ab^	0.07 ± 0.00 ^a^	0.06 ± 0.00 ^abc^	0.06 ± 0.00 ^bc^	0.06 ± 0.01 ^abc^	0.05 ± 0.00 ^c^	0.06 ± 0.00 ^bc^
	2,6 dimethyl–4–heptanone	C108838	C9H18O	142.2	967.8	1.32856	0.36 ± 0.00 ^d^	0.39 ± 0.01 ^d^	0.56 ± 0.00 ^bc^	0.52 ± 0.00 ^c^	0.72 ± 0.07 a	0.54 ± 0.01 ^c^	0.60 ± 0.01 ^b^
	3–pentanone–M	C96220	C5H10O	86.1	693.2	1.10836	0.41 ± 0.00 ^a^	0.44 ± 0.00 ^a^	0.30 ± 0.00 ^b^	0.24 ± 0.00 ^c^	0.19 ± 0.01 ^d^	0.13 ± 0.00 ^e^	0.07 ± 0.00 ^f^
	3–pentanone–D	C96220	C5H10O	86.1	692.6	1.35946	1.01 ± 0.01 ^b^	1.39 ± 0.01 ^a^	1.27 ± 0.00 ^a^	1.47 ± 0.01 ^a^	0.74 ± 0.30 ^c^	0.57 ± 0.01 ^c^	0.71 ± 0.01 ^c^
	2–pentanone	C107879	C5H10O	86.1	676.5	1.37584	3.85 ± 0.00 ^b^	2.62 ± 0.01 ^cd^	3.92 ± 0.00 ^b^	6.04 ± 0.00 ^a^	2.78 ± 0.03 ^c^	1.96 ± 0.01 ^d^	2.00 ± 0.01 ^d^
	2–hexanone	C591786	C6H12O	100.2	779.5	1.18752	0.31 ± 0.01 ^a^	0.17 ± 0.00 ^b^	0.09 ± 0.00 ^f^	0.13 ± 0.00 ^cd^	0.16 ± 0.02 ^bc^	0.11 ± 0.00 ^de^	0.04 ± 0.00 ^g^
	3–hydroxybutan–2–one–M	C513860	C4H8O2	88.1	712.8	1.05627	1.00 ± 0.04 ^d^	1.40 ± 0.01 ^c^	1.29 ± 0.05 ^c^	1.31 ± 0.01 ^c^	2.07 ± 0.01 ^a^	1.77 ± 0.00 ^b^	0.10 ± 0.00 ^f^
	3–hydroxybutan–2–one–D	C513860	C4H8O2	88.1	713.2	1.33282	10.44 ± 0.01 ^a^	2.67 ± 0.01 ^ef^	4.56 ± 0.00 ^d^	1.74 ± 0.00 ^f^	3.34 ± 0.20 ^de^	3.60 ± 0.01 ^de^	7.35 ± 0.02 ^c^
	2,3–hexanedione–M	C3848246	C6H10O2	114.1	778.3	1.09198	0.15 ± 0.01 ^c^	0.12 ± 0.00 ^d^	0.15 ± 0.00 c	0.16 ± 0.00 ^c^	0.31 ± 0.01 ^a^	0.27 ± 0.00 ^b^	0.07 ± 0.00 ^e^
	2,3–hexanedione–D	C3848246	C6H10O2	114.1	772.7	1.35966	0.30 ± 0.01 ^bcd^	0.24 ± 0.01 ^cd^	0.35 ± 0.02 ^bc^	0.18 ± 0.01 ^d^	0.39 ± 0.13 ^bc^	0.44 ± 0.00 ^b^	0.80 ± 0.01 ^a^
	(E)–3–penten–2–one–M	C3102338	C5H8O	84.1	731.8	1.35105	0.03 ± 0.02 ^bc^	0.02 ± 0.01 ^c^	0.04 ± 0.01 ^b^	0.04 ± 0.00 ^b^	0.10 ± 0.00 ^a^	0.11 ± 0.00 ^a^	0.01 ± 0.00 ^c^
	(E)–3–penten–2–one–D	C3102338	C5H8O	84.1	734.2	1.09263	0.72 ± 0.03 ^d^	0.17 ± 0.01 ^f^	1.05 ± 0.02 ^bc^	0.14 ± 0.02 ^e^	0.96 ± 0.08 ^cd^	1.23 ± 0.04 ^b^	2.14 ± 0.01 ^a^
	2–heptanone–M	C110430	C7H14O	114.2	891.8	1.25627	0.79 ± 0.01 ^a^	0.50 ± 0.01 ^b^	0.18 ± 0.02 ^e^	0.34 ± 0.01 ^d^	0.54 ± 0.02 ^b^	0.48 ± 0.01 ^bc^	0.27 ± 0.01 ^d^
	2–heptanone–D	C110430	C7H14O	114.2	890.1	1.63201	1.63 ± 0.00 ^d^	1.32 ± 0.01 ^e^	2.27 ± 0.00 ^ab^	2.34 ± 0.06 ^a^	2.14 ± 0.10 ^abc^	1.61 ± 0.07 ^d^	1.92 ± 0.07 ^c^
	1–octen–3–one–M	C4312996	C8H14O	126.2	979.4	1.27092	0.11 ± 0.00 ^de^	0.13 ± 0.00 ^cd^	0.12 ± 0.01 ^de^	0.11 ± 0.00 ^e^	0.14 ± 0.02 ^ab^	0.14 ± 0.02 ^bc^	0.16 ± 0.02 ^a^
	1–octen–3–one–D	C4312996	C8H14O	126.2	978.7	1.68592	0.04 ± 0.00 ^b^	0.04 ± 0.00 ^b^	0.05 ± 0.00 ^ab^	0.06 ± 0.00 ^a^	0.04 ± 0.00 ^b^	0.04 ± 0.01 ^b^	0.05 ± 0.01 ^ab^
	2–butanone–M	C78933	C4H8O	72.1	565.1	1.06071	0.46 ± 0.02 ^b^	0.46 ± 0.01 ^b^	0.38 ± 0.02 ^d^	0.26 ± 0.01 ^e^	0.62 ± 0.03 ^a^	0.43 ± 0.03 ^bc^	0.27 ± 0.01 ^e^
	2–butanone–D	C78933	C4H8O	72.1	566.8	1.24553	20.44 ± 0.01 ^b^	14.45 ± 0.07 ^e^	16.38 ± 0.04 ^d^	23.23 ± 0.06 ^a^	12.95 ± 0.07 ^f^	13.80 ± 0.01 ^ef^	14.81 ± 0.01 ^e^
	2–ethyl–1–hexanol	C104767	C8H18O	130.2	1026.9	1.26013	0.12 ± 0.00 ^d^	0.13 ± 0.01 ^d^	0.21 ± 0.00 ^b^	0.17 ± 0.00 ^c^	0.20 ± 0.07 ^b^	0.22 ± 0.00 ^b^	0.29 ± 0.00 ^a^
	Oct–1–en–3–ol–M	C3391864	C8H16O	128.2	986	1.15975	3.61 ± 0.01 ^d^	4.82 ± 0.01 ^ab^	4.99 ± 0.02 ^a^	4.12 ± 0.03 ^c^	3.06 ± 0.10 ^e^	2.87 ± 0.01 ^e^	4.09 ± 0.01 ^c^
	Oct–1–en–3–ol–D	C3391864	C8H16O	128.2	979.4	1.60406	0.09 ± 0.00 ^de^	0.15 ± 0.01 ^b^	0.20 ± 0.00 ^a^	0.15 ± 0.00 ^b^	0.11 ± 0.01 ^de^	0.09 ± 0.01 ^e^	0.14 ± 0.01 ^bc^
	n–hexanol–M	C111273	C6H14O	102.2	861.9	1.32497	1.22 ± 0.01 ^bc^	1.20 ± 0.03 ^c^	1.05 ± 0.00 ^d^	1.26 ± 0.02 ^bc^	1.56 ± 0.02 ^a^	1.28 ± 0.00 ^b^	0.86 ± 0.01 ^f^
	n–hexanol–D	C111273	C6H14O	102.2	861.9	1.6385	2.39 ± 0.00 ^e^	3.18 ± 0.01 ^cd^	4.80 ± 0.03 ^a^	4.69 ± 0.00 ^a^	4.02 ± 0.01 ^b^	3.29 ± 0.09 ^c^	2.89 ± 0.03 ^d^
	(E)–2–hexen–1–ol–M	C928950	C6H12O	100.2	846	1.18518	0.22 ± 0.00 ^d^	0.10 ± 0.02 ^g^	0.17 ± 0.00 ^e^	0.19 ± 0.00 ^def^	0.50 ± 0.01 ^a^	0.43 ± 0.01 ^b^	0.33 ± 0.02 ^c^
	(E)–2–hexen–1–ol–D	C928950	C6H12O	100.2	844	1.51752	0.04 ± 0.00 ^d^	0.15 ± 0.00 ^b^	0.10 ± 0.00 ^c^	0.10 ± 0.00 ^c^	0.15 ± 0.04 ^b^	0.16 ± 0.00 ^b^	0.40 ± 0.01 ^a^
	3–methylbutan–1–ol	C123513	C5H12O	88.1	727.3	1.48806	0.80 ± 0.00 ^c^	2.22 ± 0.02 ^a^	0.95 ± 0.01 ^bc^	1.16 ± 0.00 ^b^	0.45 ± 0.01 ^d^	0.19 ± 0.00 ^e^	0.09 ± 0.00 ^e^
	1–butanol–M	C71363	C4H10O	74.1	654.3	1.1823	0.52 ± 0.02 ^b^	0.38 ± 0.04 ^c^	0.36 ± 0.04 ^c^	0.61 ± 0.03 ^a^	0.36 ± 0.05 ^c^	0.27 ± 0.02 ^d^	0.09 ± 0.01 ^f^
	1–butanol–D	C71363	C4H10O	74.1	653	1.37683	0.18 ± 0.00 ^a^	0.12 ± 0.00 ^c^	0.06 ± 0.01 ^f^	0.06 ± 0.00 ^f^	0.10 ± 0.00 ^de^	0.10 ± 0.00 ^de^	0.14 ± 0.00 ^b^
	Methanethiol	C74931	CH4S	48.1	441.9	1.0483	5.77 ± 0.01 ^ab^	5.97 ± 0.04 ^a^	5.16 ± 0.07 ^d^	5.21 ± 0.03 ^d^	2.41 ± 0.05 ^g^	2.80 ± 0.00 ^f^	4.74 ± 0.00 ^e^
	2–furanmethanethiol	C98022	C5H6OS	114.2	912.8	1.11099	0.06 ± 0.00 ^a^	0.05 ± 0.00 ^ab^	0.03 ± 0.00 ^c^	0.05 ± 0.00 ^b^	0.01 ± 0.00 ^d^	0.01 ± 0.00 ^d^	0.02 ± 0.00 ^d^
	Pentan–1–ol–M	C71410	C5H12O	88.1	762.4	1.25392	1.13 ± 0.00 ^d^	1.26 ± 0.01 ^c^	1.28 ± 0.00 ^c^	1.30 ± 0.01 ^c^	1.96 ± 0.09 ^a^	1.69 ± 0.04 ^b^	0.64 ± 0.01 ^e^
	Pentan–1–ol–D	C71410	C5H12O	88.1	764.4	1.51286	0.54 ± 0.01 ^f^	1.41 ± 0.01 ^cd^	2.43 ± 0.02 ^a^	2.22 ± 0.01 ^b^	1.46 ± 0.20 ^c^	1.50 ± 0.01 ^c^	0.59 ± 0.01 ^f^
Hydrocarbon	Styrene	C100425	C8H8	104.2	887.9	1.41987	0.23 ± 0.00 ^bc^	0.39 ± 0.01 ^a^	0.25 ± 0.01 ^bc^	0.27 ± 0.01 ^b^	0.10 ± 0.0 ^d^	0.08 ± 0.00 ^d^	0.10 ± 0.00 ^d^
	Toluene	C108883	C7H8	92.1	760.5	1.01492	0.11 ± 0.00 ^b^	0.19 ± 0.00 ^c^	0.13 ± 0.01 ^d^	0.16 ± 0.00 ^c^	0.27 ± 0.03 ^a^	0.21 ± 0.01 ^b^	0.08 ± 0.00 ^f^
Acids	Hexanoic acid	C142621	C6H12O2	116.2	994.1	1.30503	0.69 ± 0.01 ^b^	0.80 ± 0.00 ^a^	0.57 ± 0.01 ^c^	0.42 ± 0.01 ^d^	0.31 ± 0.01 ^e^	0.26 ± 0.01 ^ef^	0.23 ± 0.02 ^f^
Furan	2–pentyl furan	C3777693	C9H14O	138.2	994.1	1.2552	0.28 ± 0.01 ^e^	0.59 ± 0.01 ^a^	0.55 ± 0.03 ^ab^	0.44 ± 0.05 ^c^	0.32 ± 0.03 ^d^	0.36 ± 0.01 ^d^	0.53 ± 0.01 ^b^

Note: Different lowercase letters represent significant differences between different soaking treatments (*p* < 0.05). Abbreviations: DT, drift time; MW, molecular weight; RI, retention index.

**Table 4 foods-12-03877-t004:** Fatty acid content in beef liver and beef liver steak under different soaking conditions (/%).

Serial Number	Fatty Acid Composition	A	B–30	B–60	B–90	C–30	C–60	C–90	KA	KB–30	KB–60	KB–90	KC–30	KC–60	KC–90
1	C12:0	0.46 ± 0.02 ^d^	0.53 ± 0.03 ^c^	0.65 ± 0.03 ^b^	0.59 ± 0.01 ^c^	0.69 ± 0.02 ^b^	0.73 ± 0.07 ^a^	0.66 ± 0.03 ^b^	1.03 ± 0.07 ^cd^	1.09 ± 0.06 ^cd^	1.16 ± 0.04 ^c^	1.19 ± 0.07 ^c^	1.18 ± 0.04 ^c^	1.55 ± 0.06 ^a^	1.31 ± 0.05 ^b^
2	C13:0	0.04 ± 0.01 ^b^	0.05 ± 0.01 ^ab^	0.05 ± 0.01 ^ab^	0.04 ± 0.01 ^b^	0.06 ± 0.01 ^a^	0.06 ± 0.01 ^a^	0.06 ± 0.02 ^a^	0.06 ± 0.01 ^ab^	0.08 ± 0.02 ^a^	0.07 ± 0.03 ^a^	0.06 ± 0.02 ^ab^	0.05 ± 0.02 ^b^	0.07 ± 0.02 ^a^	0.05 ± 0.01 ^b^
3	C14:0	1.10 ± 0.02 ^d^	1.30 ± 0.03 ^c^	1.43 ± 0.01 ^b^	1.48 ± 0.02 ^b^	1.58 ± 0.03 ^a^	1.65 ± 0.10 ^a^	1.21 ± 0.03 ^c^	1.35 ± 0.10 ^c^	1.52 ± 0.20 ^a^	1.58 ± 0.23 ^a^	1.49 ± 0.03 ^b^	1.45 ± 0.05 ^ab^	1.65 ± 0.03 ^a^	1.53 ± 0.03 ^a^
4	C14:1	0.23 ± 0.01 ^c^	0.26 ± 0.03 ^bc^	0.31 ± 0.01 ^b^	0.29 ± 0.01 ^b^	0.32 ± 0.03 ^b^	0.36 ± 0.02 ^a^	0.19 ± 0.03 ^d^	10.57 ± 0.02 ^b^	10.61 ± 0.03 ^b^	10.63 ± 0.05 ^ab^	10.59 ± 0.02 ^b^	10.72 ± 0.06 ^a^	10.65 ± 0.03 ^ab^	10.79 ± 0.03 ^a^
5	C15:0	0.14 ± 0.03 ^e^	0.20 ± 0.02 ^d^	0.28 ± 0.03 ^c^	0.26 ± 0.04 ^c^	0.32 ± 0.01 ^b^	0.35 ± 0.03 ^a^	0.18 ± 0.03 ^d^	0.43 ± 0.03 ^a^	0.39 ± 0.05 ^b^	0.46 ± 0.06 ^a^	0.52 ± 0.03 ^a^	0.38 ± 0.02 ^b^	0.35 ± 0.03 ^b^	0.42 ± 0.02 ^a^
6	C16:0	14.26 ± 0.05 ^c^	15.28 ± 0.12 ^c^	16.23 ± 0.08 ^b^	16.05 ± 0.06 ^b^	16.85 ± 0.09 ^b^	17.07 ± 0.25 ^a^	15.00 ± 0.08 ^c^	13.07 ± 0.25 ^a^	12.43 ± 0.18 ^b^	12.84 ± 0.24 ^b^	13.02 ± 0.07 ^a^	12.85 ± 0.25 ^b^	13.25 ± 0.05 ^a^	12.25 ± 0.06 ^b^
7	C16:1	2.05 ± 0.02 ^c^	2.65 ± 0.04 ^b^	3.21 ± 0.05 ^a^	2.92 ± 0.06 ^ab^	3.02 ± 0.08 ^ab^	3.28 ± 0.12 ^a^	2.43 ± 0.06 ^b^	13.18 ± 0.12 ^c^	13.25 ± 0.30 ^c^	13.31 ± 0.19 ^bc^	13.63 ± 0.37 ^b^	13.45 ± 0.04 ^b^	14.65 ± 0.03 ^a^	13.24 ± 0.03 ^b^
8	C17:0	1.87 ± 0.02 ^b^	1.75 ± 0.03 ^b^	2.11 ± 0.03 ^a^	2.08 ± 0.06 ^a^	2.23 ± 0.02 ^a^	2.39 ± 0.01 ^a^	1.65 ± 0.03 ^bc^	1.89 ± 0.01 ^c^	1.73 ± 0.02 ^c^	2.02 ± 0.03 ^b^	2.13 ± 0.05 ^b^	2.29 ± 0.02 ^a^	2.25 ± 0.04 ^a^	2.08 ± 0.01 ^a^
9	C17:1	0.36 ± 0.03 ^b^	0.42 ± 0.02 ^a^	0.52 ± 0.02 ^a^	0.49 ± 0.03 ^a^	0.59 ± 0.03 ^a^	0.66 ± 0.09 ^a^	0.46 ± 0.03 ^a^	10.66 ± 0.09 ^b^	10.73 ± 0.04 ^a^	10.76 ± 0.06 ^b^	10.82 ± 0.04 ^a^	10.72 ± 0.03 ^a^	10.89 ± 0.05 ^a^	10.66 ± 0.04 ^b^
10	C18:0	16.65 ± 0.04 ^d^	17.46 ± 0.08 ^c^	18.5 ± 0.09 ^bc^	18.89 ± 0.03 ^b^	19.24 ± 0.12 ^b^	20.08 ± 1.31 ^b^	26.85 ± 0.03 ^a^	23.08 ± 0.31 ^a^	22.43 ± 0.08 ^ab^	23.05 ± 0.04 ^a^	24.38 ± 0.08 ^a^	18.98 ± 0.05 ^b^	14.16 ± 0.08 ^c^	18.48 ± 0.05 ^b^
11	C18:1	14.35 ± 0.02 ^c^	15.29 ± 0.07 ^c^	15.59 ± 0.21 ^c^	16.02 ± 0.03 ^b^	16.65 ± 0.19 ^b^	17.29 ± 0.35 ^b^	24.15 ± 0.06 ^a^	15.29 ± 0.35 ^d^	16.02 ± 0.19 ^cd^	16.34 ± 0.23 ^c^	16.85 ± 0.76 ^c^	26.89 ± 0.03 ^c^	27.52 ± 0.05 ^b^	28.79 ± 0.04 ^a^
12	C18:2	16.25 ± 0.08 ^c^	17.62 ± 0.06 ^b^	18.79 ± 0.09 ^a^	18.66 ± 0.05 ^a^	19.58 ± 0.19 ^a^	20.22 ± 0.36 ^a^	17.68 ± 0.05 ^b^	22.22 ± 0.36 ^d^	23.35 ± 0.04 ^c^	24.11 ± 0.09 ^c^	25.36 ± 0.05 ^b^	25.51 ± 0.05 ^b^	27.23 ± 0.08 ^a^	23.21 ± 0.05 ^c^
13	C20:0	0.57 ± 0.01 ^b^	0.62 ± 0.03 ^ab^	0.79 ± 0.05 ^ab^	0.68 ± 0.06 ^ab^	0.87 ± 0.06 ^a^	0.84 ± 0.02 ^a^	0.51 ± 0.01 ^b^	10.54 ± 0.02 ^a^	10.38 ± 0.03 ^ab^	10.42 ± 0.01 ^a^	10.46 ± 0.03 ^a^	10.29 ± 0.02 ^bc^	10.34 ± 0.01 ^a^	10.19 ± 0.01 ^c^
14	C20:1	0.05 ± 0.01 ^b^	0.06 ± 0.03 ^b^	0.07 ± 0.02 ^a^	0.08 ± 0.05 ^a^	0.09 ± 0.03 ^a^	0.08 ± 0.01 ^a^	0.06 ± 0.01 ^b^	10.08 ± 0.01 ^a^	10.10 ± 0.02 ^a^	10.16 ± 0.03 ^a^	10.18 ± 0.04 ^a^	10.17 ± 0.02 ^a^	10.25 ± 0.01 ^a^	10.15 ± 0.01 ^a^
15	C20:2	0.63 ± 0.03 ^b^	0.56 ± 0.03 ^c^	0.67 ± 0.01 ^ab^	0.64 ± 0.02 ^b^	0.70 ± 0.02 ^b^	0.82 ± 0.03 ^a^	0.60 ± 0.03 ^b^	10.62 ± 0.03 ^a^	10.54 ± 0.01 ^a^	10.35 ± 0.02 ^b^	10.47 ± 0.06 ^ab^	10.53 ± 0.05 ^a^	10.42 ± 0.02 ^ab^	10.35 ± 0.01 ^b^
16	C20:3n6	7.65 ± 0.01 ^c^	8.16 ± 0.08 ^b^	9.16 ± 0.03 ^a^	8.85 ± 0.07 ^ab^	9.05 ± 0.05 ^b^	9.62 ± 0.17 ^a^	7.89 ± 0.05 ^c^	19.82 ± 0.17 ^a^	10.02 ± 0.03 ^c^	10.25 ± 0.06 ^c^	10.17 ± 0.01 ^c^	11.48 ± 0.05 ^bc^	12.63 ± 0.12 ^b^	10.37 ± 0.04 ^c^
	SFA	45.09 ^a^	45.99 ^b^	44.10 ^d^	45.07 ^d^	41.18 ^d^	43.77 ^c^	46.12 ^b^	42.45 ^c^	44.35 ^a^	44.09 ^a^	43.95 ^c^	43.53 ^b^	43.36 ^b^	44.48 ^a^
	USFA	54.91 ^d^	54.01 ^b^	55.32 ^a^	54.93 ^a^	58.82 ^a^	56.23 ^b^	53.88 ^c^	53.55 ^c^	55.65 ^bc^	55.91 ^b^	56.07 ^a^	56.47 ^a^	56.64 ^b^	55.52 ^b^
	MUFAs	20.01 ^ab^	18.68 ^bc^	19.60 ^b^	19.80 ^b^	20.67 ^ab^	21.67 ^a^	22.21 ^a^	32.44 ^c^	34.01 ^b^	34.87 ^b^	36.18 ^a^	35.79 ^a^	33.53 ^c^	34.08 ^b^
	PUFAs	34.90 ^c^	36.33 ^b^	38.62 ^b^	40.13 ^a^	38.15 ^b^	30.56 ^d^	31.67 ^d^	21.11 ^d^	20.64 ^d^	21.04 d	21.89 ^c^	21.68 ^c^	22.11 ^ab^	23.48 ^a^

Note: Different lowercase letters represent significant differences between different soaking treatments (*p* < 0.05).

## Data Availability

The data generated from this study are clearly presented and discussed in the manuscript. Data is contained within the article.

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
