# Peer review of "Effects of Salt Soaking Treatment on the Deodorization of Beef Liver and the Flavor Formation of Beef Liver Steak"

_foods, 2023, doi:10.3390/foods12203877_

Round 1

Reviewer 1 Report

Comments and Suggestions for Authors

The article studied the effect of salt soaking of beef liver on its deodorization and changing the taste of beef liver steak. The work was carried out according to the classical canons of Food Science, which establishes the connection between food processing and organoleptic properties. The strength of the study is the use of the GC–IMS technique, which provides great opportunities for characterizing volatile compounds.

I think that the article can be improved taking into account the following comments and suggestions:

- When the authors present the results of GC–IMS analysis, the authors present difference spectra. It is necessary to clarify what is meant by the term “difference”. Since this method is relatively new, readers would be interested to know more about this method. It is better that the authors give a large-scale spectrum for one sample as an example. For example, you can give a spectrum in three-dimensional space in the coordinates: retention time, drift time, intensity (not color, but intensity values). Shown in Fig. 3 spectra are very small and nothing is visible.

- It is necessary to provide an explanation regarding the designations RI and Dt (Table 2 and 3). In what units was Dt measured? How many significant digits should be given for RI and Dt. Six significant digits are too many (Table 2 and 3).

- The inserts in Figures 4 and 5 are difficult to see. The drawings need to be made larger.

Author Response

Dear editor and reviewers:

We would like to thank the editors and reviewers for your valuable time reviewing our paper. We would like to express our sincere gratitude to the reviewers for their feedback on our paper, "Effects of salt soaking treatment on deodorization of beef liver and flavor formation of beef liver steak". These valuable comments helped us further revise and improve the paper. We have carefully reviewed these comments and revised them one by one. Without affecting the overall content and framework of the paper, we did our best to embellish the language in the manuscript. If there are other revisions that can be made, we would very much like to do so. We sincerely thank the editors and reviewers for your enthusiastic work and hope for your approval!

In the revised manuscript, changes are shown in red (changes made in response to issues raised by editors and reviewers) and blue (corrected for English grammatical and spelling errors and embellished corresponding statements). Detailed responses to the reviewers' comments are listed below, together with the exact line numbers of the changes noted in the responses. Once again, we thank the editor and reviewers for your positive comments and valuable input to improve the quality of our manuscript. The revised version is attached for your review.

I look forward to your reply.

Thank you and best regards.

Yours sincerely

Yufeng Duan

Reviewer 2 Report

Comments and Suggestions for Authors

The manuscript foods-2634935 entitled " Effects of salt soaking treatment on deodorization of beef liver and flavor formation of beef liver steak

The manuscript is well-written, and informations and the contents of the manuscript are of great value. However, there are some areas that require further improvement to enhance the overall quality of the paper

1)     In its current state, the level of English throughout your manuscript does not meet the journal's desired standard. There are many badly worded/constructed sentences. Extensive rephasing and paraphrasing led to a change in the actual meaning of the sentences. Difficult to understand.

2)     Figures (2-6). Please improve figure quality by increasing size and DPI. The figures in current form are difficult to understand. You can increase size by separating them.

3)     The abstract, in the present form, is not adequate. Please add specific results here not generalized. Additionally, (i) the authors must state the revised justification in the abstract to support the study; (ii) Conclusion in the abstract is too general, authors shall state the important finding in this study before going into the general application

4)     The conclusion section is redundant with results already summarized and analyzed. In the present form same the abstract of the paper. This section is not adequate rewrite please.

5)     Why pH was recorded?

6)     In conclusions, the authors discussed the effect of addition of salt on lipid oxidation. Why they did not measure TBARS?

“The fatty acid results show that the addition of salt can affect the lipid oxidation”

7)     Keywords: Please replace the keywords which are already appeared in the main title of the manuscript. No need to repeat here like Beef liver, Beef liver steak, Deodorization, Flavor.

8)     Line 27: Please delete the word of waste and write as Meat by–products

Comments on the Quality of English Language

 In its current state, the level of English throughout your manuscript does not meet the journal's desired standard. There are many badly worded/constructed sentences. Extensive rephasing and paraphrasing led to a change in the actual meaning of the sentences. Difficult to understand.

Author Response

(The authors gave the same response as above.)

Round 2

Reviewer 2 Report

Comments and Suggestions for Authors

I am satisfied with the authors response.